# One Subgoal at a Time: Zero-Shot Generalization to Arbitrary Linear Temporal Logic Requirements in Multi-Task Reinforcement Learning

**Zijian Guo[1], İlker Işık[2], H. M. Sabbir Ahmad[1], Wenchao Li[1,2]**
[1]Division of Systems Engineering, Boston University
[2]Department of Electrical and Computer Engineering, Boston University
{zjguo, iilker, sabbir92, wenchao}@bu.edu

## Abstract

Generalizing to complex and temporally extended task objectives and safety constraints remains a critical challenge in reinforcement learning (RL). Linear temporal logic (LTL) offers a unified formalism to specify such requirements, yet existing methods are limited in their abilities to handle nested long-horizon tasks and safety constraints, and cannot identify situations when a subgoal is not satisfiable and an alternative should be sought. In this paper, we introduce GenZ-LTL, a method that enables zero-shot generalization to arbitrary LTL specifications. GenZ-LTL leverages the structure of Büchi automata to decompose an LTL task specification into sequences of reach-avoid subgoals. Contrary to the current state-of-the-art method that conditions on subgoal sequences, we show that it is more effective to achieve zero-shot generalization by solving these reach-avoid problems *one subgoal at a time* through proper safe RL formulations. In addition, we introduce a novel subgoal-induced observation reduction technique that can mitigate the exponential complexity of subgoal-state combinations under realistic assumptions. Empirical results show that GenZ-LTL substantially outperforms existing methods in zero-shot generalization to unseen LTL specifications. The code is available at https://github.com/BU-DEPEND-Lab/GenZ-LTL.

## 1 Introduction

Generalization is a critical aspect of reinforcement learning (RL), which aims to develop policies that are capable of adapting to novel and unseen tasks [38, 48, 21] and varying safety constraints [7, 17]. Various methods have been proposed and show promising results in applications such as autonomous driving [49, 36], robotics [27, 33, 28], and healthcare [8, 41]. However, most existing approaches fall short in handling complex and temporally extended task objectives and safety constraints. For example, an autonomous vehicle may be instructed to reach a sequence of destinations for passenger pick-up and drop-off while adhering to traffic rules such as speed limits and yielding at intersections.

Temporal logic provides a principled framework for specifying system behaviors over time. Linear temporal logic (LTL) [39], in particular, has been considered extensively in single-task and multi-task RL settings [20, 29, 51]. LTL has well-defined semantics and a compositional syntax, which facilitates the unambiguous specification of task objectives and safety constraints. However, existing methods often fail to account for the potential conflicting objectives within a specification [50, 40, 23]. For example, an autonomous vehicle navigating between multiple destinations aims to minimize travel time, but adhering to speed limits and other safety constraints may slow the vehicle down. In such cases, the vehicle must prioritize satisfying safety requirements over optimizing for travel time.

39th Conference on Neural Information Processing Systems (NeurIPS 2025).

In this paper, we propose a novel framework, called GenZ-LTL, for learning RL policies that generalize in a zero-shot manner to satisfy arbitrary LTL specifications, including both finite-horizon and infinite-horizon tasks. We adopt the subgoal decomposition approach, which exploits the structure of the equivalent Büchi automaton to decompose an LTL specification into individual subgoals, each comprising a *reach* component that defines task progression and an *avoid* component that encodes safety constraints. We show that training a goal-conditioned policy to achieve the subgoals *one at a time*, which is often perceived as myopic by existing literature, is more effective in general than conditioning the policy on subgoal sequences [23] or the entire automaton [54]. This somewhat surprising result stems from two key innovations in this paper: a proper safe RL formulation with *state-wise* constraints for solving the individual reach-avoid subgoals, and a novel *subgoal-induced observation reduction* technique to address the exponential complexity of observation-subgoal combinations. The former builds on recent advances of Hamilton-Jacobi reachability in RL [12] whereas the latter exploits symmetry in the state observations to extract information only relevant to the immediate subgoal. To zero-shot generalize to arbitrary LTL specifications at test time, we use the equivalent Büchi automata as well as the learned value functions to guide the subgoal selection. Our main contributions are summarized as follows.

- We propose a novel approach for learning RL policies that can satisfy arbitrary LTL specifications in a zero-shot manner by sequentially completing individual subgoals. To our knowledge, this is the first safe RL formulation with state-wise constraints for LTL task satisfaction.

- We introduce a subgoal-induced observation reduction technique to mitigate the combinatorial explosion of observation-subgoal pairs and enable efficient generalization, along with a timeout mechanism that handles subgoal switching and unsatisfiable subgoals.

- We conduct comprehensive experiments across a set of navigation-style environments and diverse task specifications, demonstrating that our method consistently outperforms state-of-the-art baselines in zero-shot generalization to arbitrary LTL specifications.

## 2 Related Work

**Specification-guided and goal-conditioned RL.** Various methods have been proposed to guide RL agents by extending traditional goal-conditioned RL [35] to temporal-logic goals. Early works focused on learning policies for a single and fixed specification, including methods based on LTL specifications and automaton-like models [20, 18, 51, 46, 45, 3], and methods leveraging quantitative semantics of specifications [32, 26, 44, 16]. More recently, the importance of adapting to changing conditions in real-world applications has brought increased attention to zero-shot generalization to new specifications. Existing approaches can be broadly categorized into subgoal decomposition [30, 31, 34, 40, 23] and direct specification/automata encoding [50, 54, 55]. Our work follows the direction of subgoal decomposition, but shows that it is more effective to solve the subgoals one at a time through proper safe RL formulations and subgoal-induced observation reduction.

**RL with safety constraints.** Various methods have been proposed to solve RL problems with safety constraints, such as policy optimization-based methods [59, 56, 24, 53], Gaussian processes-based approaches [52, 60], and control theory-based methods, including Hamilton-Jacobian (HJ) reachability [10, 57, 47, 13, 61] and control barrier functions [14, 6]. We point the readers to [15] for a comprehensive survey of safe RL techniques. In the context of formal specification-guided RL, the interplay between task progression and safety constraints remains underexplored. Most existing methods simply assign a positive reward when a specification or a subgoal is satisfied, and a negative reward at episode termination if unsatisfied. This approach is ineffective in achieving safety [1, 9, 4], especially when the task progression objective and the safety requirements exert conflicting influences on policy learning. In this paper, we show how to prioritize safety satisfaction properly by incorporating Hamilton-Jacobi reachability constraints.

## 3 Preliminaries

**Reinforcement learning.** RL agents learn policies by interacting with environments, which is usually modeled as a Markov Decision Process (MDP), defined by the tuple $\mathcal{M} := (\mathcal{S}, \mathcal{A}, P, r, \gamma, d_0)$, where $\mathcal{S}$ is the state space and $\mathcal{A}$ is the action space, $P : \mathcal{S} \times \mathcal{A} \times \mathcal{S} \mapsto [0, 1]$ defines the transition dynamics, $r : \mathcal{S} \times \mathcal{A} \mapsto \mathbb{R}$ is the reward function, $\gamma \in [0, 1)$ is the discount factor, and $d_0 \in \Delta(\mathcal{S})$ is the initial state distribution. Let $\pi : \mathcal{S} \times \mathcal{A} \mapsto [0, 1]$ denote a policy and $\tau = \{s_t, a_t, r_t\}_{t=0}^{\infty}$ denote a trajectory

where $r_t = r(s_t, a_t)$. The value function under policy $\pi$ is $V_r^\pi(s) = \mathbb{E}_{\tau \sim \pi, s_0 = s}[\sum_{t=0}^\infty \gamma^t r_t]$ and the corresponding state-action value function is $Q_r^\pi(s, a) = \mathbb{E}_{\tau \sim \pi, s_0 = s, a_0 = a}[\sum_{t=1}^\infty \gamma^t r_t]$. An additional constraint violation function $h : \mathcal{S} \mapsto \mathbb{R}$ can be used to model state-wise safety constraints, such that a state $s$ is considered safe if $h(s) \leq 0$ and unsafe if $h(s) > 0$. The reachability value function $V_h^\pi(s)$ is defined as $V_h^\pi(s) := \max_{t \in \mathbb{N}} h(s), s_0 = s, a \sim \pi$ that captures the worst-case constraint violation under policy $\pi$ from state $s$. The largest feasible set contains the largest set of states from which the agent can reach a goal safely: $\mathcal{S}_f := \{s \mid \exists \pi, V_h^\pi(s) \leq 0\}$. To handle such constraints, Hamilton-Jacobi (HJ) reachability has been incorporated into RL [12] by simultaneously learning the reachability value function to estimate the feasible set and learning the policy:

$$\max_\pi \mathbb{E}_{s \sim d_0} [V_r^\pi(s) \cdot \mathbb{1}[s \in \mathcal{S}_f] - V_h^\pi(s) \cdot \mathbb{1}[s \notin \mathcal{S}_f]] \quad \text{s.t. } V_h^\pi(s) \leq 0, \forall s \in \mathcal{S}_f \cap \mathcal{S}_0$$

where $\mathcal{S}_0 := \{s \mid s \sim d_0\}$ is the set of initial states. Intuitively, the goal is to maximize expected return and ensure safety for initial states within the largest feasible set, while minimizing the reachability value function for states outside it, where constraint violations are inevitable.

**Linear Temporal Logic.** LTL [39] is a formal language for specifying temporal properties of a system. LTL formulas are built from a set of Boolean operators – negation ($\neg$), conjunction ($\wedge$), disjunction ($\vee$), and temporal operators – "until" ($\mathsf{U}$), "eventually" ($\mathsf{F}$), and "always" ($\mathsf{G}$). Given a finite set of atomic propositions $AP$ and $\boldsymbol{a} \in AP$, the syntax of LTL is defined recursively as:

$$\varphi := \boldsymbol{a} \mid \neg\varphi \mid \varphi_1 \wedge \varphi_2 \mid \varphi_1 \vee \varphi_2 \mid \mathsf{F}\,\varphi \mid \mathsf{G}\,\varphi \mid \varphi_1 \,\mathsf{U}\, \varphi_2$$

Intuitively, the formula $\varphi_1 \,\mathsf{U}\, \varphi_2$ holds if $\varphi_1$ is satisfied at all time steps prior to the first occurrence of $\varphi_2$. The operator $\mathsf{F}\,\varphi$ holds if $\varphi$ is satisfied at some future time, while $\mathsf{G}\,\varphi$ holds if $\varphi$ is satisfied from the current step onward. To relate LTL to MDPs, we assume a known labeling function $L : \mathcal{S} \mapsto 2^{AP}$ that maps each state to the set of true atomic propositions. The satisfaction probability of $\varphi$ under a policy $\pi$ is $\Pr(\pi \models \varphi) = \mathbb{E}_{\tau \sim \pi}[\mathbb{1}[\tau \models \varphi]]$, where $\tau \models \varphi$ denotes that the trace of the trajectory, $\mathrm{Tr}(\tau) = L(s_0)L(s_1)\dots$, satisfies the formula $\varphi$, and $\mathbb{1}$ is the indicator function. We are interested in training a policy that maximizes this probability for an arbitrary $\varphi$.

**Büchi automata and subgoals.** To enable algorithmic reasoning over LTL specifications, we use Büchi automata [5] as a formal representation of temporal-logic formulas. Given an LTL formula $\varphi$, the corresponding (non-deterministic) Büchi automaton is defined by a tuple $\mathcal{B}_\varphi := (\mathcal{Q}, \Sigma, \delta, \mathcal{F}, q_0)$ where $\mathcal{Q}$ is the finite set of automaton states, $\Sigma = 2^{AP}$ the finite alphabet, $\delta \subseteq \mathcal{Q} \times \Sigma \times \mathcal{Q}$ the transition relation, $\mathcal{F} \subseteq \mathcal{Q}$ the set of accepting automaton states, and $q_0$ the initial state. An infinite word $\omega = \alpha_0 \alpha_1 \dots \in \Sigma^\omega$ induces a run $r = q_0 q_1 \dots$ over $\omega$ if for every $i \geq 0$, $(q_i, \alpha_i, q_{i+1}) \in \delta$. A run $r$ is accepting if it visits an accepting state infinitely often. A product MDP $\mathcal{M}^\varphi$ [11] synchronizes a given MDP and Büchi automaton with a new state space $\mathcal{S}^\varphi = \mathcal{S} \times \mathcal{Q}$ and a transition function $\mathcal{P}^\varphi((s', q') \mid (s, q), a)$ that equals $\mathcal{P}(s' \mid s, a)$ if $a \in \mathcal{A}$ and $(q, L(s), q') \in \delta$, and 0 otherwise. In practice, rather than explicitly building the product MDP, one can simply update the current automaton state $q$ with the propositions $L(s)$ observed at each time step. Given a state $q$, a *reach* subgoal is defined as an $\alpha^+ \in \Sigma$ such that $(q, \alpha^+, q') \in \delta$ and $q$ and $q'$ are consecutive automaton states along some accepting run, and $q' \neq q$ if $q \notin \mathcal{F}$. Conversely, an *avoid* subgoal is defined as an $\alpha^- \in \Sigma$ such that $(q, \alpha^-, q') \in \delta$ and $q$ and $q'$ are not consecutive automaton states along any accepting run. Intuitively, a reach subgoal indicates progression towards satisfying the LTL specification, and an avoid subgoal indicates a way that leads to violation of the specification.

## 4 Method

In this section, we present our proposed method, as illustrated in Figure 1. During training, we begin by sampling subgoals $\sigma = (\alpha^+, A^-)$, where $\alpha^+$ is a single reach subgoal and $A^-$ is a (possibly empty) set of avoid subgoals (and $\alpha^+ \notin A^-$). Given the current state $s$ and subgoal $\sigma$, we apply subgoal-induced observation reduction (details in Section 4.2), when applicable, to obtain a processed state $s^\sigma$. We then train a general reach-avoid subgoal-conditioned policy along with its corresponding value functions. During testing, given a target LTL task specification $\varphi$, we first construct an equivalent Büchi automaton, and then extract a set of reach-avoid subgoals $\{\sigma_i\}_{i=1}^n$ based on the current MDP and automaton state. For each candidate subgoal $\sigma_i$, we compute $s^{\sigma_i}$ and evaluate the value functions to select the optimal subgoal $\sigma^*$. The agent then generates actions $a$ according to $\pi(a|s^{\sigma^*})$. In Section 4.1, we describe how to construct reach-avoid subgoals from a Büchi automaton. In Section 4.2, we introduce a subgoal-induced observation reduction technique to

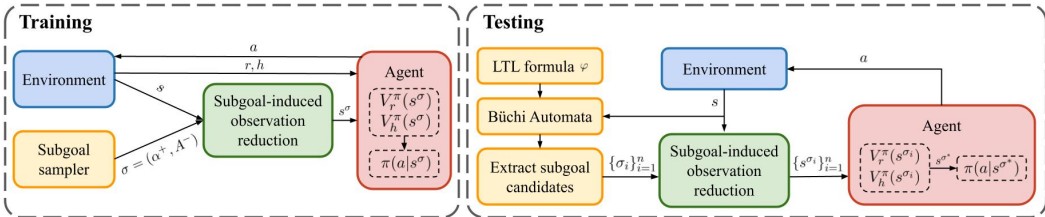

Figure 1: Overview of GenZ-LTL. During training, we sample reach-avoid subgoals $\sigma = (\alpha^+, A^-)$ and apply a subgoal-induced observation reduction to learn a subgoal-conditioned policy and value functions. At test time, given a target LTL formula $\varphi$, we construct the corresponding Büchi automaton and identify candidate subgoals based on the current automaton state. The optimal subgoal $\sigma^*$ is selected using the learned value functions, and the policy generates actions conditioned on $\sigma^*$.

mitigate the exponential complexity arising from the combination of subgoals and MDP states. In Section 4.3, we detail the learning of value functions and policies under reachability constraints to comply with the safety constraints ($A^-$) while optimizing for the task progression objective ($\alpha^+$).

## 4.1 Reach-Avoid Subgoal Construction from Büchi Automaton

Büchi automata can be represented as directed state-transition graphs [40], and an accepted run can be viewed as having a lasso structure with a cycle containing an accepting state and a prefix path that leads to the cycle. Similar to existing approaches [40, 23], we use depth-first search (DFS) over the state-transition graph to enumerate all possible lasso structures starting from a given state $q$. From these lasso paths, we extract the set of reach subgoals, denoted by $A^+$, following the definition in Section 3. Similarly, we extract the set of avoid subgoals $A^-$ by considering transitions to a state that is different from $q$ and not along any of the lasso paths. Each pair $(\alpha^+ \in A^+, A^-)$ then becomes a candidate reach-avoid subgoal. To satisfy a subgoal $(\alpha^+, A^-)$, the agent must visit a state $s$ such that $L(s) = \alpha^+$, while avoiding any states labeled with propositions in $A^-$. For example, for $(A^+ = \{\{a, b\}, \{c\}\}, A^- = \{\{d\}, \{e\}\})$, it forms subgoals $\sigma_1 = (\{a \wedge b\}, A^-)$ and $\sigma_2 = (\{c\}, A^-)$, and the agent can choose to reach a state labeled with $a \wedge b$ or a state labeled with $c$, while avoiding states labeled with $d$ and $e$ in both cases. While our method follows a similar subgoal decomposition approach as prior works, the way that we solve the reach-avoid subgoals is significantly different. In fact, a straightforward way is to encode a reach-avoid subgoal as a bitvector of dimension $|AP| + 2^{|AP|}$, where the first $|AP|$ bits encodes the reach subgoal, and the rest of the bits encode the avoid subgoal.

## 4.2 Subgoal-Induced Observation Reduction

Given a reach-avoid subgoal, our next step is to train a subgoal-conditioned policy based on the MDP state or observation of the MDP state, and the subgoal. However, notice that the number of possible subgoals grows exponentially with the number of atomic propositions. Additionally, in order for the agent to complete a task in such a complex environment, it needs to be able to make observation of every state-subgoal combination, i.e. being able to determine the label $L(s)$ of a state $s$ that it observes (possibly indirectly through an observation function $o$). Otherwise, task completion cannot be determined in general. For instance, if the task requires the agent to reach a 'green' region, then an agent equipped with only 'blue' sensors cannot learn to accomplish this task if the green and blue regions are placed randomly in the environment. Thus, the number of observations that the policy needs to condition on also grows exponentially with $|AP|$, and directly learning a goal-conditioned policy this way quickly becomes intractable as $|AP|$ grows. While various dimensionality reduction techniques have been proposed in the RL literature [2, 19, 58], they are based on abstraction or bisimulation of the MDP states, and do not consider state-subgoal combinations.

In this paper, we introduce a novel subgoal-induced observation reduction technique that can circumvent this exponential complexity. The key observation is that, ultimately, each subgoal reduces to a reach-avoid problem, where there is some state that we want to reach while avoiding some other states. In other words, what really matters is whether a state is 'reach' or 'avoid', and not its label $L(s)$ once a subgoal is fixed. Thus, we can perform a symbolic simplification of the state/observation space

along with the subgoal while preserving information essential for solving the reach-avoid problem. Formally, for an MDP state $s$, we assume that it can be partition into a *proposition-independent* component $s_{\neg AP}$ and a *proposition-dependent* component $s_{AP}$ (e.g, the blue and green sensors). For the proposition-dependent component, we further assume that it can be broken down into individual observations each corresponding to an element in $2^{AP}$, and the observations are produced by *identical observation functions under output transformation*. We say that two observation functions (with a slight abuse of notation of $s$), $o_1(s) \in \mathbb{R}^k$ and $o_2(s) \in \mathbb{R}^k$ are identical under an output transformation $g$ if $\forall s, o_1(s) = g(o_2(s))$. For instance, suppose $o_1(s) \in \mathbb{R}^{360}$ is a LiDAR sensor for sensing the shortest distance to a blue state along each of its 360 beams, and $o_2(s) \in \mathbb{R}^{360}$ is the same type of LiDAR sensor for sensing the shortest distance to a green state but rotated $180°$, then $g$ is the $180°$ rotation (a permutation of the beam indices). Then, we can reduce the proposition-dependent component (of size $k \times 2^{|AP|}$) and the encoding of a reach-avoid subgoal (of size $(|AP| + 2^{|AP|})$ into just two observations each of size $k$, where the first observation is an observation corresponding to the reach subgoal and the second observation is a *fused* observation of the observations corresponding to the avoid subgoals. For simplicity of deriving a fusion operator that preserves task-relevant semantics such as distances to avoid zones, we assume that the $o(s)$ are coordinate-wise monotonic. The actual fusion operator $f$ is environment/sensor-specific – in the LiDAR example, as shown in Figure 2, it will be an elementwise $\min$ over $\mathbb{R}^k$. Implementation details are provided in Appendix B. Empirically, we demonstrate the benefits of this reduction, when it is applicable, in Sections 5.3 and Apendix C.

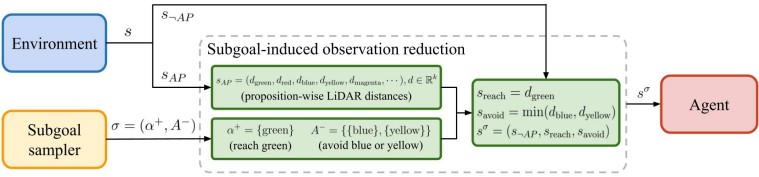

Figure 2: Illustration of the subgoal-induced observation reduction for LiDAR observations. The environment provides the raw state consisting of a proposition-independent component $s_{\neg AP}$ and a proposition-dependent component $s_{AP}$. Given a subgoal $\sigma$, $s_{AP}$ is reduced to two vectors: a reach observation and a fused avoid observation. Together with the $s_{\neg AP}$, these form the agent's input.

## 4.3 Goal-Conditioned Policy Learning with Reachability Constraints

A key difference between our approach and existing subgoal-based approaches lies in the treatment of the avoid subgoal. We treat an avoid subgoal as a *hard* constraint, since whenever a violation occurs, satisfying the LTL formula becomes impossible. For example, consider the specification $\neg a \ \mathsf{U} \ b$, where $a$ and $b$ are atomic propositions; this requires the agent to avoid reaching any state labeled with $a$ until a state labeled with $b$ is reached. If $a$ is encountered before $b$, the whole trajectory can no longer satisfy the specification. To enforce these hard constraints, we leverage the notion of HJ reachability as introduced in Section 3 and formulate our RL objective as follows:

$$\max_{\pi} \mathbb{E}_{s \sim d_0, \sigma \sim \text{Unif}(\xi)} \left[ V_r^{\pi}(s^{\sigma}) \cdot \mathbb{1}[s^{\sigma} \in \mathcal{S}_f^{\sigma}] - V_h^{\pi}(s^{\sigma}) \cdot \mathbb{1}[s^{\sigma} \notin \mathcal{S}_f^{\sigma}] \right]$$
$$\text{s.t.} \ V_h^{\pi}(s^{\sigma}) \leq 0, \forall s^{\sigma} \in \mathcal{S}_f^{\sigma} \cap \mathcal{S}_0^{\sigma}, \forall \sigma \sim \text{Unif}(\xi) \tag{1}$$

where $\xi$ is the set of all possible subgoals (details on its construction are provided in Appendix B), $s^{\sigma} := f(s, \sigma)$ is the processed state if the observation reduction is applied, and $s^{\sigma} := (s, \sigma)$ otherwise, $\mathcal{S}_f^{\sigma} := \{s^{\sigma} \mid \sigma, \exists \pi, V_h^{\pi}(s^{\sigma}) \leq 0\}$ denotes the largest subgoal-dependent feasible set, $\mathcal{S}_0^{\sigma} := \{s^{\sigma} \mid \sigma, s \sim d_0\}$ denotes the set of processed initial states given a subgoal $\sigma$, and $V_h^{\pi}(s^{\sigma})$ is the reachability value function of a policy $\pi$ starting from a processed state $s^{\sigma}$ with the current subgoal $\sigma$. We use $\pi$ as a shorthand for the policy $\pi(a \mid s^{\sigma})$. However, since $\mathcal{S}_f^{\sigma}$ is unknown when learning policies, directly solving Eq. (1) is not possible. Inspired by previous methods [1, 57], we convert the objective into the following constrained policy optimization problem:

$$\pi_{k+1} = \arg\max_{\pi} \mathbb{E}_{\sigma \sim \text{Unif}(\xi), s \sim d^{\pi_k}, a \sim \pi_k} \left[ \frac{\pi}{\pi_k} A_r^{\pi_k}(s^{\sigma}, a) \right]$$
$$\text{s.t.} \quad \mathbb{E}_{\sigma \sim \text{Unif}(\xi), s \sim d^{\pi_k}} [\mathcal{D}_{KL}(\pi, \pi_k)] \leq \epsilon, \tag{2}$$
$$\mathbb{E}_{\sigma \sim \text{Unif}(\xi), s \sim d^{\pi_k}, a \sim \pi_k} \left[ (1 - \gamma) J_h(\pi_k) + \frac{\pi}{\pi_k} A_h^{\pi_k}(s^{\sigma}, a) \right] \leq 0$$

where $A_r^\pi(s^\sigma, a) := Q_r^\pi(s^\sigma, a) - V_r^\pi(s^\sigma)$ and $A_h^\pi(s^\sigma, a) := Q_h^\pi(s^\sigma, a) - V_h^\pi(s^\sigma)$ are the advantages functions with $Q_h^\pi(s^\sigma, a) := \max_{t \in \mathbb{N}} h(s_t^\sigma), s_0 = s^\sigma, a_0 = a, a_t \sim \pi$ and $J_h(\pi) := \max_{t \in \mathbb{N}} h(\cdot), a \sim \pi$ is the maximum safety violation for a trajectory under policy $\pi$. By maximizing the discounted cumulative reward, the policy is optimized to effectively satisfy the reach subgoal, while the safety constraints ensures that the avoid subgoals are satisfied, Then we incorporate the clipped surrogate objective [43] to handle the trust-region constraint and a *state-dependent* Lagrangian multiplier [57] to enforce the state-wise safety constraint:

$$\min_{\lambda \geq 0} \max_\pi \mathbb{E}_{\sigma \sim \mathrm{Unif}(\xi), s \sim d^{\pi_k}, a \sim \pi} \left[ \min(\frac{\pi}{\pi_k} A_r^{\pi_k}(s^\sigma, a), \ \mathrm{clip}(\frac{\pi}{\pi_k}, 1 - \epsilon, 1 + \epsilon) A_r^{\pi_k}(s^\sigma, a)) \right.$$
$$\left. - \lambda(s^\sigma)((1 - \gamma) J_h(s^\sigma) + \frac{\pi}{\pi_k} A_h^{\pi_k}(s^\sigma, a)) \right] \tag{3}$$

The overall method is summarized in Algorithm 1. The training process of our method is simple compared to previous methods that sample full LTL specifications [50, 40] or employ curriculum learning that gradually exposes the policy to more challenging reach-avoid sequences [23]. We sample one subgoal at a time, apply observation reduction to obtain a reduced state when applicable, and execute the policy to collect data. When the subgoal is satisfied, a new one is sampled, and this cycle continues until the episode terminates due to reaching the maximum length or a subgoal violation. During evaluation, given an LTL formula $\varphi$, we convert it to a Büchi automaton and track the current automaton state $q$. We identify the set of candidate subgoals $\{\sigma_i\}_{i=1}^n$ that can advance toward satisfying $\varphi$. The subgoal that achieves the best trade-off between task progression and satisfying safety constraint is selected as $\sigma^* = \arg\max_\sigma V_r(s^\sigma) - \lambda(s^\sigma) V_h(s^\sigma)$. The agent takes actions based on the selected subgoal and updates it upon each transition to a new automaton state.

**Subgoal switching.** In general, we cannot determine *a priori* whether a selected subgoal is satisfiable in the environment. To handle this issue, we introduce a mechanism to identify unsatisfiable subgoals and allow the agent to switch to alternative ones. The mechanism is triggered if the current subgoal is not satisfied within a timeout threshold, which is set to $(1 + \epsilon_{\mathrm{scale}}) \cdot \mu_{\mathrm{subgoal}}$, where $\mu_{\mathrm{subgoal}}$ denotes the maximum number of steps required to complete a subgoal after policy convergence during training, and $\epsilon_{\mathrm{scale}}$ is a scaling factor. When a reach subgoal times out, the best option from the remaining reach subgoals in $A^+$ is selected. This procedure is sound because the avoid subgoals are respected during the attempt for the prior reach subgoal(s). Appendix B.2 provides further details.

*Remark.* If there is no more subgoal remaining, our method returns failure. However, this does not necessarily mean that the specification is unsatisfiable. It is possible that the specification is satisfiable but the greedy nature of solving the subgoals sequentially prevents us from finding a solution. In general, unless the environment MDP is known, we cannot determine *a priori* if the LTL specification is satisfiable or not. If the agent is allowed to reset in the same environment, then one can enumerate the possible subgoal sequences from the initial state of the automaton.

## 5 Experiments

In this section, we evaluate our method GenZ-LTL in multiple environments and across a wide range of LTL specifications of varying complexity to answer the following questions: **Q1:** How well can GenZ-LTL zero-shot generalize to arbitrary LTL specifications? **Q2:** How does GenZ-LTL perform under increasing specification complexity and environment complexity? **Q3:** What are the individual impacts of the proposed observation reduction and safe RL approach? Additional evaluations and results in are provided in Appendix C.

**Environments.** Our environments include the `LetterWorld` environment [50], a $7 \times 7$ discrete grid world with discrete actions where letters are placed at randomly sampled positions, and the `ZoneEnv` environment [50], a high-dimensional environment with lidar observations, where a robotic agent with a continuous action space must navigate between randomly positioned colored regions. We exclude the `FlatWorld` environment [23, 51] from our experiments, as its configuration is fixed and the state space includes only the agent's $(x, y)$-position, without any information about the environment state. We also consider a variant of `ZoneEnv` that contains overlapping regions which is by design more challenging than `FlatWorld`. Additional details are provided in Appendix A.1.

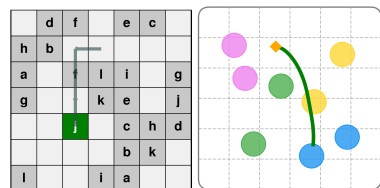

Figure 3: Environment illustrations of `LetterWorld` and `ZoneEnv` with example trajectories for ¬l U j and ¬green U blue, respectively.

**Baselines.** We compare GenZ-LTL against the following baselines, all of which train a goal-conditioned policy of some sort: 1) LTL2Action [50], which applies a technique called LTL progression to dynamically track the parts of a specification that remain to be satisfied, and use a graph neural network to encode the specification; 2) GCRL-LTL [40], which uses a heuristic to identify the optimal subgoal sequence; 3) DeepLTL [23], which learns policies conditioned on embeddings of subgoal sequences; 4) RAD-embeddings [54], which pre-trains embeddings of compositions of deterministic finite automata (DFAs) and learns policies conditioned on these embeddings. While these baselines consider safety, as the LTL specification itself can involve the notion of safety, they do not model safety constraints explicitly in their RL formulation. Further details of the implementation are provided in Appendix B.

**LTL specifications.** We consider a wide variety of LTL specifications with varying complexity, including (1) sequential reach-only specifications of the form $\mathsf{F}\,(\alpha_1 \wedge (\mathsf{F}\,\alpha_2 \wedge (\ldots \wedge \mathsf{F}\,\alpha_n))$; (2) sequential reach-avoid specifications of the form $\neg\alpha_1 \,\mathsf{U}\, (\alpha_2 \wedge (\neg\alpha_3 \,\mathsf{U}\, (\alpha_4 \wedge \ldots \wedge (\neg\alpha_{n-1} \,\mathsf{U}\, \alpha_n)))$, and (3) complex specifications that combine reach-only and reach-avoid properties. We also conduct a separate evaluation on infinite-horizon tasks that span a diverse set of LTL objectives, including safety, liveness, and their combinations. The specifications used for evaluation are listed in Table 1 and a full set of specifications is provided in Appendix A.2, with trajectory visualizations in Appendix D. In this section, all specifications are satisfiable.

Table 1: LTL specifications used for evaluation. For specifications in `ZoneEnv`, b, g, m, and y denote blue, green, `magenta`, and yellow, respectively. Results are shown in Table 2 and 3.

| | | LetterWorld | | ZoneEnv |
|---|---|---|---|---|
| Finite-horizon | $\varphi_1$ | $\mathsf{F}\,(a \wedge (\neg b\,\mathsf{U}\,c)) \wedge \mathsf{F}\,d$ | $\varphi_9$ | $(\mathsf{F}\,b) \wedge (\neg b\,\mathsf{U}\,(g \wedge \mathsf{F}\,y))$ |
| | $\varphi_2$ | $(\mathsf{F}\,d) \wedge (\neg f\,\mathsf{U}\,(d \wedge \mathsf{F}\,b))$ | $\varphi_{10}$ | $\neg(m \vee y)\,\mathsf{U}\,(b \wedge \mathsf{F}\,g)$ |
| | $\varphi_3$ | $\neg a\,\mathsf{U}\,(b \wedge (\neg c\,\mathsf{U}\,(d \wedge (\neg e\,\mathsf{U}\,f))))$ | $\varphi_{11}$ | $\neg g\,\mathsf{U}\,((b \vee m) \wedge (\neg g\,\mathsf{U}\,y))$ |
| | $\varphi_4$ | $(a \vee b \vee c \vee d \Rightarrow \mathsf{F}\,(e \wedge \mathsf{F}\,(f \wedge \mathsf{F}\,g)))\,\mathsf{U}\,(h \wedge \mathsf{F}\,i)$ | $\varphi_{12}$ | $(g \vee b \Rightarrow (\neg y\,\mathsf{U}\,m))\,\mathsf{U}\,y$ |
| | $\varphi_5$ | $\mathsf{F}\,(d \wedge (\neg(a \vee b)\,\mathsf{U}\,(b \wedge (\neg e\,\mathsf{U}\,c)))) \wedge \mathsf{F}\,(\neg(f \vee g \vee h)\,\mathsf{U}\,a)$ | $\varphi_{13}$ | $\mathsf{F}\,(g \wedge (\neg(b \vee y)\,\mathsf{U}\,(y \wedge (\neg m\,\mathsf{U}\,b)))) \wedge \mathsf{F}\,(\neg g\,\mathsf{U}\,y)$ |
| | $\varphi_6$ | $\mathsf{F}\,((k \wedge ((\neg b \vee c)\,\mathsf{U}\,f)) \wedge (\neg(a \vee e \vee h)\,\mathsf{U}\,g)) \wedge \mathsf{F}\,d$ | $\varphi_{14}$ | $\mathsf{F}\,((b \vee g) \wedge (\neg y\,\mathsf{U}\,(b \wedge (\neg(g \vee m)\,\mathsf{U}\,m)))) \wedge \mathsf{F}\,(y \wedge (\neg b\,\mathsf{U}\,g))$ |
| | $\varphi_7$ | $\neg(j \vee b \vee d)\,\mathsf{U}\,(a \wedge (\neg c\,\mathsf{U}\,(f \wedge \mathsf{F}\,(g \wedge (\neg d\,\mathsf{U}\,e)))))$ | $\varphi_{15}$ | $\neg(m \vee y)\,\mathsf{U}\,(b \wedge (\neg g\,\mathsf{U}\,(y \wedge \mathsf{F}\,(g \wedge (\neg b\,\mathsf{U}\,m)))))$ |
| | $\varphi_8$ | $\neg(f \vee g)\,\mathsf{U}\,(a \wedge (\neg b\,\mathsf{U}\,c) \wedge \mathsf{F}\,(d \wedge (\neg e\,\mathsf{U}\,f)))$ | $\varphi_{16}$ | $\mathsf{F}\,(b \wedge (\neg y\,\mathsf{U}\,(g \wedge \mathsf{F}\,(y \wedge (\neg(m \vee g)\,\mathsf{U}\,b)))))$ |
| Infinite-horizon | $\psi_1$ | $\mathsf{GF}\,(e \wedge (\neg a\,\mathsf{U}\,f)) \wedge \mathsf{G}\,\neg(c \vee d)$ | $\psi_4$ | $\mathsf{GF}\,b \wedge \mathsf{GF}\,g \wedge \mathsf{G}\,\neg(y \vee m)$ |
| | $\psi_2$ | $\mathsf{GF}\,a \wedge \mathsf{GF}\,b \wedge \mathsf{GF}\,c \wedge \mathsf{G}\,\neg(e \vee f \vee i)$ | $\psi_5$ | $\mathsf{GF}\,b \wedge \mathsf{GF}\,y \wedge \mathsf{GF}\,g \wedge \mathsf{G}\,\neg m$ |
| | $\psi_3$ | $\mathsf{GF}\,c \wedge \mathsf{GF}\,a \wedge \mathsf{GF}\,(e \wedge (\neg f\,\mathsf{U}\,g)) \wedge \mathsf{GF}\,k \wedge \mathsf{G}\,\neg(i \vee j)$ | $\psi_6$ | $\mathsf{FG}\,y \wedge \mathsf{G}\,\neg(g \vee b \vee m)$ |

**Evaluation metrics.** For fair comparisons, we follow the standard practice of executing a deterministic policy by taking the mean of the action distribution [22]. For each LTL specification, we report the following rates: success rate $\eta_s$, defined as the ratio of trajectories that satisfy the specification; violation rate $\eta_v$, defined as the ratio of trajectories that violate it; and other rate $\eta_o = 1 - (\eta_s + \eta_v)$. We also report the average number of steps $\mu$ to satisfy the specification among successful trajectories as a secondary metric, as step count is less important when the success rate is low.

## 5.1 Can GenZ-LTL zero-shot generalize to arbitrary LTL specifications (Q1)?

**Finite-horizon tasks.** We first evaluate finite-horizon tasks, i.e. tasks that can be satisfied with finite-length trajectories. All the specifications considered are complex LTL formulas with nested temporal and Boolean operators (details in Table 1). The evaluation results are shown in Table 2. Our method consistently outperforms the baselines in terms of both success and violation rates. LTL2Action encodes the full structure of the current LTL specification, making it less effective at adapting to out-of-distribution (OOD) specifications at test time. GCRL-LTL learns a goal-conditioned policy but does not model safety constraints explicitly, limiting its ability to enforce constraint satisfaction. DeepLTL relies on curriculum training which samples random reach-avoid subgoal sequences only up to a certain length. When evaluating specifications $\varphi_9$–$\varphi_{12}$, although the specifications themselves are unseen, the underlying subgoal sequences are likely to have been encountered during training, leading to better efficiency, i.e., smaller $\mu$. However, for specifications with longer subgoal sequences, e.g, $\varphi_{13}$–$\varphi_{16}$, which are OOD with respect to the training sequences, DeepLTL exhibits reduced performance. RAD-embeddings pre-trains the automaton embeddings by sampling from a fixed distribution. As a result, it also struggles with the issue of OOD generalization when faced with unseen

Table 2: Evaluation results for success rate $\eta_s$, violation rate $\eta_v$, and average steps $\mu$ to satisfy the complex specifications listed in Table 1. Specifications $\varphi_1$–$\varphi_8$ are evaluated in `LetterWorld`, and $\varphi_9$–$\varphi_{16}$ in `ZoneEnv`. $\varphi_1$-$\varphi_4$ and $\varphi_9$-$\varphi_{12}$ are from [23] and we construct more complex $\varphi_5$-$\varphi_8$ and $\varphi_{13}$-$\varphi_{16}$. ↑: higher is better; ↓: lower is better. **Bold**: the best performance for each metric. Each value is averaged over 5 seeds, with 100 trajectories per seed.

| | | LTL2Action | | | GCRL-LTL | | | DeepLTL | | | RAD-embeddings | | | GenZ-LTL(ours) | | |
|---|---|---|---|---|---|---|---|---|---|---|---|---|---|---|---|---|
| | | $\eta_s\uparrow$ | $\eta_v\downarrow$ | $\mu\downarrow$ | $\eta_s\uparrow$ | $\eta_v\downarrow$ | $\mu\downarrow$ | $\eta_s\uparrow$ | $\eta_v\downarrow$ | $\mu\downarrow$ | $\eta_s\uparrow$ | $\eta_v\downarrow$ | $\mu\downarrow$ | $\eta_s\uparrow$ | $\eta_v\downarrow$ | $\mu\downarrow$ |
| LetterWorld $\varphi_1$ | | $0.51_{\pm0.10}$ | $\mathbf{0.00_{\pm0.00}}$ | $29.03_{\pm3.08}$ | $0.87_{\pm0.04}$ | $0.02_{\pm0.02}$ | $15.37_{\pm0.39}$ | $0.87_{\pm0.02}$ | $\mathbf{0.00_{\pm0.00}}$ | $8.12_{\pm0.21}$ | $0.96_{\pm0.04}$ | $\mathbf{0.00_{\pm0.00}}$ | $18.71_{\pm2.06}$ | $\mathbf{0.98_{\pm0.02}}$ | $0.00_{\pm0.00}$ | $\mathbf{7.38_{\pm0.13}}$ |
| $\varphi_2$ | | $0.66_{\pm0.16}$ | $0.14_{\pm0.11}$ | $23.25_{\pm4.93}$ | $0.90_{\pm0.03}$ | $\mathbf{0.00_{\pm0.00}}$ | $9.26_{\pm0.75}$ | $0.91_{\pm0.02}$ | $0.01_{\pm0.01}$ | $5.88_{\pm0.12}$ | $0.90_{\pm0.07}$ | $0.07_{\pm0.04}$ | $13.64_{\pm2.71}$ | $\mathbf{1.00_{\pm0.00}}$ | $0.00_{\pm0.00}$ | $\mathbf{5.48_{\pm0.09}}$ |
| $\varphi_3$ | | $0.75_{\pm0.07}$ | $0.14_{\pm0.04}$ | $24.60_{\pm4.38}$ | $0.72_{\pm0.06}$ | $0.09_{\pm0.07}$ | $14.02_{\pm1.09}$ | $0.81_{\pm0.02}$ | $0.01_{\pm0.00}$ | $9.00_{\pm0.14}$ | $0.86_{\pm0.05}$ | $0.08_{\pm0.03}$ | $20.02_{\pm2.05}$ | $\mathbf{0.96_{\pm0.01}}$ | $0.00_{\pm0.00}$ | $\mathbf{8.61_{\pm0.46}}$ |
| $\varphi_4$ | | $0.56_{\pm0.19}$ | $\mathbf{0.00_{\pm0.00}}$ | $29.69_{\pm8.50}$ | $\mathbf{0.99_{\pm0.00}}$ | $0.00_{\pm0.00}$ | $25.54_{\pm0.84}$ | $0.89_{\pm0.02}$ | $0.00_{\pm0.00}$ | $\mathbf{7.05_{\pm0.28}}$ | $0.89_{\pm0.04}$ | $0.00_{\pm0.00}$ | $18.78_{\pm2.82}$ | $0.98_{\pm0.01}$ | $0.00_{\pm0.00}$ | $7.41_{\pm0.37}$ |
| Ave. | | $0.62_{\pm0.16}$ | $0.07_{\pm0.09}$ | $26.64_{\pm5.87}$ | $0.87_{\pm0.11}$ | $0.03_{\pm0.05}$ | $16.05_{\pm6.13}$ | $0.87_{\pm0.04}$ | $\mathbf{0.00_{\pm0.01}}$ | $7.51_{\pm1.21}$ | $0.90_{\pm0.06}$ | $0.04_{\pm0.04}$ | $17.79_{\pm3.37}$ | $\mathbf{0.98_{\pm0.02}}$ | $0.00_{\pm0.00}$ | $\mathbf{7.22_{\pm1.18}}$ |
| $\varphi_5$ | | $0.30_{\pm0.13}$ | $\mathbf{0.00_{\pm0.00}}$ | $39.12_{\pm2.08}$ | $0.67_{\pm0.07}$ | $0.07_{\pm0.05}$ | $20.53_{\pm0.49}$ | $0.78_{\pm0.05}$ | $0.00_{\pm0.00}$ | $11.24_{\pm0.39}$ | $0.88_{\pm0.05}$ | $0.00_{\pm0.00}$ | $26.02_{\pm1.62}$ | $\mathbf{0.99_{\pm0.00}}$ | $0.00_{\pm0.00}$ | $\mathbf{10.60_{\pm0.12}}$ |
| $\varphi_6$ | | $0.18_{\pm0.10}$ | $\mathbf{0.00_{\pm0.00}}$ | $28.40_{\pm6.15}$ | $0.70_{\pm0.08}$ | $0.15_{\pm0.05}$ | $14.69_{\pm0.75}$ | $0.81_{\pm0.02}$ | $0.00_{\pm0.00}$ | $8.23_{\pm0.18}$ | $0.90_{\pm0.04}$ | $0.00_{\pm0.00}$ | $19.48_{\pm1.96}$ | $\mathbf{0.95_{\pm0.01}}$ | $0.00_{\pm0.00}$ | $\mathbf{7.48_{\pm0.06}}$ |
| $\varphi_7$ | | $0.29_{\pm0.16}$ | $0.33_{\pm0.20}$ | $40.71_{\pm1.43}$ | $0.59_{\pm0.05}$ | $0.14_{\pm0.07}$ | $19.52_{\pm1.25}$ | $0.71_{\pm0.02}$ | $0.03_{\pm0.01}$ | $11.82_{\pm0.39}$ | $0.73_{\pm0.05}$ | $0.17_{\pm0.05}$ | $26.07_{\pm3.73}$ | $\mathbf{0.94_{\pm0.01}}$ | $0.00_{\pm0.00}$ | $\mathbf{10.97_{\pm0.08}}$ |
| $\varphi_8$ | | $0.20_{\pm0.08}$ | $0.47_{\pm0.22}$ | $37.50_{\pm4.16}$ | $0.63_{\pm0.09}$ | $0.09_{\pm0.04}$ | $20.10_{\pm0.82}$ | $0.75_{\pm0.02}$ | $0.02_{\pm0.01}$ | $11.20_{\pm0.25}$ | $0.76_{\pm0.11}$ | $0.13_{\pm0.05}$ | $25.60_{\pm2.89}$ | $\mathbf{0.93_{\pm0.01}}$ | $0.00_{\pm0.00}$ | $\mathbf{10.24_{\pm0.08}}$ |
| Ave. | | $0.24_{\pm0.12}$ | $0.20_{\pm0.25}$ | $36.43_{\pm6.08}$ | $0.65_{\pm0.08}$ | $0.11_{\pm0.06}$ | $18.71_{\pm2.54}$ | $0.76_{\pm0.05}$ | $0.01_{\pm0.02}$ | $10.62_{\pm1.47}$ | $0.82_{\pm0.10}$ | $0.07_{\pm0.08}$ | $24.29_{\pm3.77}$ | $\mathbf{0.95_{\pm0.03}}$ | $0.00_{\pm0.00}$ | $\mathbf{9.82_{\pm1.42}}$ |
| ZoneEnv $\varphi_9$ | | $0.13_{\pm0.08}$ | $0.18_{\pm0.25}$ | $496.09_{\pm128.52}$ | $0.88_{\pm0.05}$ | $0.03_{\pm0.00}$ | $474.41_{\pm23.42}$ | $0.89_{\pm0.03}$ | $0.04_{\pm0.00}$ | $\mathbf{327.80_{\pm23.78}}$ | $0.90_{\pm0.07}$ | $0.08_{\pm0.07}$ | $423.11_{\pm51.08}$ | $\mathbf{0.99_{\pm0.01}}$ | $0.01_{\pm0.01}$ | $380.15_{\pm5.17}$ |
| $\varphi_{10}$ | | $0.39_{\pm0.32}$ | $0.38_{\pm0.24}$ | $421.46_{\pm58.76}$ | $0.88_{\pm0.03}$ | $0.06_{\pm0.01}$ | $303.43_{\pm4.46}$ | $0.88_{\pm0.03}$ | $0.09_{\pm0.01}$ | $\mathbf{221.94_{\pm23.26}}$ | $0.92_{\pm0.02}$ | $0.06_{\pm0.02}$ | $301.21_{\pm101.99}$ | $\mathbf{0.97_{\pm0.01}}$ | $0.02_{\pm0.02}$ | $253.44_{\pm10.22}$ |
| $\varphi_{11}$ | | $0.82_{\pm0.15}$ | $0.08_{\pm0.03}$ | $284.41_{\pm95.62}$ | $0.86_{\pm0.04}$ | $0.06_{\pm0.03}$ | $306.94_{\pm11.26}$ | $0.91_{\pm0.06}$ | $0.04_{\pm0.02}$ | $\mathbf{215.42_{\pm15.01}}$ | $0.93_{\pm0.06}$ | $0.04_{\pm0.02}$ | $247.83_{\pm42.98}$ | $\mathbf{0.99_{\pm0.01}}$ | $0.01_{\pm0.01}$ | $249.57_{\pm11.85}$ |
| $\varphi_{12}$ | | $0.94_{\pm0.03}$ | $0.02_{\pm0.03}$ | $122.89_{\pm20.97}$ | $0.90_{\pm0.01}$ | $0.05_{\pm0.02}$ | $136.32_{\pm10.27}$ | $0.97_{\pm0.01}$ | $0.01_{\pm0.00}$ | $116.40_{\pm13.07}$ | $\mathbf{1.00_{\pm0.01}}$ | $0.00_{\pm0.00}$ | $\mathbf{105.97_{\pm17.31}}$ | $1.00_{\pm0.00}$ | $0.00_{\pm0.00}$ | $135.60_{\pm8.06}$ |
| Ave. | | $0.57_{\pm0.37}$ | $0.17_{\pm0.21}$ | $331.21_{\pm165.88}$ | $0.88_{\pm0.04}$ | $0.05_{\pm0.02}$ | $305.28_{\pm123.33}$ | $0.91_{\pm0.05}$ | $0.04_{\pm0.03}$ | $220.39_{\pm78.77}$ | $0.94_{\pm0.05}$ | $0.04_{\pm0.05}$ | $269.53_{\pm129.95}$ | $\mathbf{0.99_{\pm0.01}}$ | $0.01_{\pm0.01}$ | $\mathbf{254.69_{\pm89.18}}$ |
| $\varphi_{13}$ | | $0.17_{\pm0.23}$ | $0.01_{\pm0.01}$ | $886.66_{\pm104.03}$ | $0.73_{\pm0.06}$ | $0.09_{\pm0.02}$ | $618.42_{\pm22.80}$ | $0.91_{\pm0.04}$ | $\mathbf{0.00_{\pm0.00}}$ | $508.26_{\pm54.81}$ | $0.69_{\pm0.17}$ | $0.00_{\pm0.00}$ | $664.59_{\pm29.91}$ | $\mathbf{0.98_{\pm0.02}}$ | $0.00_{\pm0.00}$ | $\mathbf{436.25_{\pm18.35}}$ |
| $\varphi_{14}$ | | $0.36_{\pm0.29}$ | $0.01_{\pm0.01}$ | $885.13_{\pm148.93}$ | $0.68_{\pm0.06}$ | $0.11_{\pm0.06}$ | $611.24_{\pm38.37}$ | $0.91_{\pm0.03}$ | $\mathbf{0.00_{\pm0.00}}$ | $473.45_{\pm52.67}$ | $0.77_{\pm0.21}$ | $0.00_{\pm0.00}$ | $628.57_{\pm22.89}$ | $\mathbf{0.98_{\pm0.01}}$ | $0.00_{\pm0.00}$ | $\mathbf{434.45_{\pm14.61}}$ |
| $\varphi_{15}$ | | $0.08_{\pm0.05}$ | $0.49_{\pm0.30}$ | $855.56_{\pm311.81}$ | $0.70_{\pm0.06}$ | $0.10_{\pm0.03}$ | $599.71_{\pm18.62}$ | $0.78_{\pm0.05}$ | $0.12_{\pm0.05}$ | $523.50_{\pm64.73}$ | $0.62_{\pm0.09}$ | $0.02_{\pm0.01}$ | $680.11_{\pm9.51}$ | $\mathbf{0.97_{\pm0.01}}$ | $0.01_{\pm0.00}$ | $\mathbf{388.01_{\pm15.47}}$ |
| $\varphi_{16}$ | | $0.03_{\pm0.04}$ | $0.02_{\pm0.04}$ | $931.19_{\pm420.81}$ | $0.68_{\pm0.09}$ | $0.08_{\pm0.04}$ | $596.29_{\pm25.87}$ | $0.89_{\pm0.10}$ | $\mathbf{0.00_{\pm0.00}}$ | $515.48_{\pm45.53}$ | $0.70_{\pm0.18}$ | $\mathbf{0.00_{\pm0.00}}$ | $616.28_{\pm59.88}$ | $\mathbf{0.99_{\pm0.01}}$ | $0.00_{\pm0.00}$ | $\mathbf{396.83_{\pm20.39}}$ |
| Ave. | | $0.12_{\pm0.18}$ | $0.17_{\pm0.28}$ | $886.39_{\pm288.60}$ | $0.70_{\pm0.07}$ | $0.09_{\pm0.04}$ | $606.42_{\pm26.76}$ | $0.87_{\pm0.08}$ | $0.03_{\pm0.06}$ | $505.17_{\pm54.03}$ | $0.70_{\pm0.15}$ | $\mathbf{0.00_{\pm0.01}}$ | $647.39_{\pm40.74}$ | $\mathbf{0.98_{\pm0.02}}$ | $0.01_{\pm0.01}$ | $\mathbf{408.71_{\pm27.47}}$ |

automata, which is reflected in its performance degradation under more complex LTL specifications: the success rate $\eta_s$ drops from 0.9 on $\varphi_{1-4}$ to 0.82 on $\varphi_{5-8}$ and 0.94 on $\varphi_{9-12}$ to 0.7 on $\varphi_{13-16}$. In contrast, our method conditions only on the current subgoal, sidestepping the issue of OOD subgoal sequences. By incorporating an HJ reachability constraint, our method improves adherence to safety constraints. Additionally, since we optimize for the discounted cumulative reward obtained upon satisfying a subgoal, efficient policies that take fewer steps are encouraged. We further evaluate our method in `ZoneEnv` with agents featuring more complex dynamics, which induce additional hard constraints such as preventing head-down falls. Our method consistently outperforms the baselines in both success and violation rates. Details of the setup and results are provided in Appendix C.

Table 3: Evaluation results for violation rate $\eta_v$ and average number of visits to accepting states $\mu_{\mathrm{acc}}$ on infinite-horizon tasks listed in Table 1. Specifications $\psi_1$–$\psi_3$ are evaluated in `LetterWorld`, and $\psi_4$–$\psi_6$ in `ZoneEnv`. ↑: higher is better; ↓: lower is better. Each value is averaged over 5 seeds, with 100 trajectories per seed. **Bold** indicates the best performance for each metric.

| Methods | Metrics | LetterWorld | | | ZoneEnv | | |
|---|---|---|---|---|---|---|---|
| | | $\psi_1$ | $\psi_2$ | $\psi_3$ | $\psi_4$ | $\psi_5$ | $\psi_6$ |
| GenZ-LTL(ours) | $\mu_{\mathrm{acc}}\uparrow$ | $\mathbf{208.95_{\pm14.39}}$ | $\mathbf{102.12_{\pm7.02}}$ | $\mathbf{55.17_{\pm1.08}}$ | $\mathbf{55.16_{\pm4.23}}$ | $\mathbf{32.75_{\pm1.20}}$ | $\mathbf{8135.67_{\pm1489.99}}$ |
| | $\eta_v\downarrow$ | $\mathbf{0.00_{\pm0.00}}$ | $\mathbf{0.00_{\pm0.00}}$ | $\mathbf{0.00_{\pm0.00}}$ | $\mathbf{0.07_{\pm0.02}}$ | $\mathbf{0.03_{\pm0.01}}$ | $\mathbf{0.03_{\pm0.02}}$ |
| DeepLTL | $\mu_{\mathrm{acc}}\uparrow$ | $142.56_{\pm22.44}$ | $48.28_{\pm12.37}$ | $19.21_{\pm4.57}$ | $30.03_{\pm13.23}$ | $15.73_{\pm4.44}$ | $7337.38_{\pm2019.56}$ |
| | $\eta_v\downarrow$ | $0.04_{\pm0.02}$ | $0.09_{\pm0.01}$ | $0.09_{\pm0.04}$ | $0.39_{\pm0.10}$ | $0.38_{\pm0.24}$ | $0.13_{\pm0.05}$ |
| GCRL-LTL | $\mu_{\mathrm{acc}}\uparrow$ | $41.98_{\pm15.80}$ | $22.77_{\pm9.50}$ | $9.53_{\pm2.28}$ | $30.00_{\pm3.72}$ | $14.61_{\pm1.62}$ | $5584.34_{\pm3180.15}$ |
| | $\eta_v\downarrow$ | $0.18_{\pm0.08}$ | $0.30_{\pm0.08}$ | $0.30_{\pm0.18}$ | $0.37_{\pm0.08}$ | $0.40_{\pm0.08}$ | $0.14_{\pm0.01}$ |

**Infinite-horizon tasks.** Next, we evaluate performance on infinite-horizon tasks, i.e. tasks that can only be satisfied with infinite-length trajectories. RAD-embeddings is excluded from this evaluation, as DFAs cannot represent such specifications. Since it is not possible to simulate infinite trajectories in these environments, we set the maximum episode length to be 10 times of the length used during training. The evaluated specifications are listed in Table 1, and results are reported in Table 3, including the average number of visits to accepting states for non-violating trajectories $\mu_{\mathrm{acc}}$ and the violation rate that is calculated based on the proportion of trajectories that violate the $\mathsf{G}\,\neg\alpha$ constraint in the evaluated specifications. For example, in the specification $\mathsf{G}\,\mathsf{F}\,\alpha_1 \wedge \mathsf{G}\,\mathsf{F}\,\alpha_2 \wedge \cdots \wedge \mathsf{G}\,\mathsf{F}\,\alpha_{n-1} \wedge \mathsf{G}\,\neg\alpha_n$, each successful cycle of $\alpha_1$-$\alpha_{n-1}$ (in any order) increases $\mu_{\mathrm{acc}}$, while any visit to $\alpha_n$ counts as a violation. In the case of $\mathsf{F}\,\mathsf{G}\,\alpha$, $\mu_{\mathrm{acc}}$ is computed as the number of consecutive steps, counted backward from the final timestep, in which $\alpha$ holds. As shown

in Table 3, our method outperforms the baselines on both metrics. By conditioning only on the current subgoal, it naturally extends to infinite-horizon tasks. The agent can follow the infinite subgoal sequence extracted from the automaton one subgoal at a time. The integration of reachability constraints further enables safe execution, whereas the baselines often struggle to satisfy safety constraints in long-horizon settings.

## 5.2 How does GenZ-LTL perform under increasing specification complexity and environment complexity (Q2)?

To further evaluate the zero-shot adaptation ability of our method, we design experiments that test performance under increasing complexity in specification and environment: (1) Subgoal chaining: we evaluate performance on reach-only and reach-avoid specifications with an increasing of sequence length of subgoals up to $n = 12$; (2) Environment complexity: we vary the region size and the number of regions associated with each atomic proposition; and (3) Unsatisfiable subgoals.

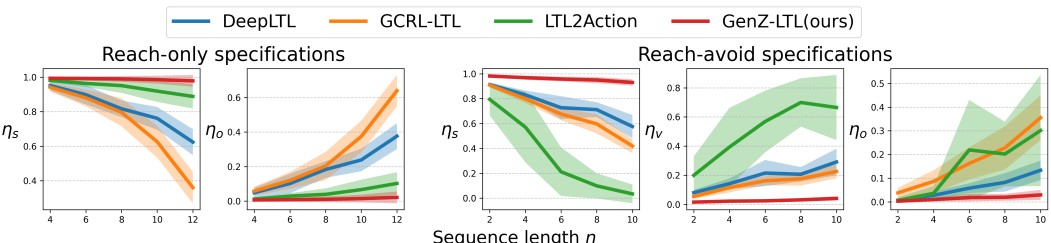

Figure 4: Performance in `ZoneEnv` under increasing subgoal sequence lengths, showing success rate $\eta_s$, violation rate $\eta_v$, and others rate $\eta_o$. We use reach-only specifications with sequence lengths $n \in \{4, 6, 8, 10, 12\}$ and reach-avoid specifications with $n \in \{2, 4, 6, 8, 10\}$. The specifications are shown in Table 4. Each value is averaged over 5 seeds, with 100 trajectories per seed.

**Subgoal chaining.** In this experiment, we increase the number of subgoals that the agent must reach sequentially. Since satisfying longer specifications naturally requires more steps, we double the maximum episode length used in Section 5.1. The results are shown in Figure 4. For the reach-only specifications, the performance of DeepLTL and GCRL-LTL degrades more rapidly as $n$ grows. In the case of DeepLTL, it suffers from OOD specifications that involve longer subgoal sequences than those used during training. GCRL-LTL performs the worst as it struggles to complete the subgoals in an efficient manner, preventing it from reaching all the subgoals within the maximum episode length. LTL2Action performs better than DeepLTL and GCRL-LTL on reach-only specifications, as the corresponding graph structures remain relatively simple compared to the complex and nested specifications in Section 5.1. Our method maintains a high success rate and a low violation rate despite the increase in sequence length, as it solves one subgoal at a time, thus avoiding the distribution shift that affects other methods and resulting in better generalization. For reach-avoid specifications, where safety satisfaction is critical, we can observe that our method achieves near-zero constraint violations compared to the baselines, which validates our safe RL approach with reachability constraints.

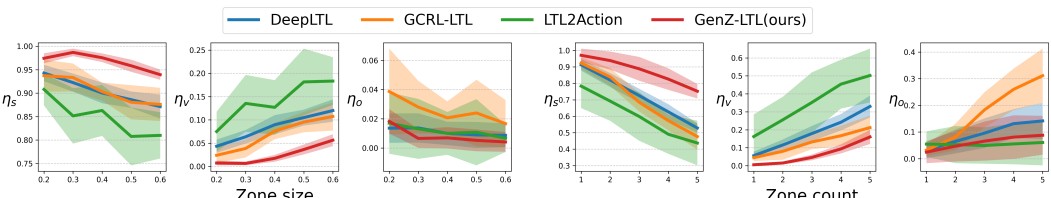

Figure 5: Performance in `ZoneEnv` under varying environment complexity, including different zone sizes and different numbers of zones per atomic proposition. Visualizations of the environment layout are provided in Figure 16. We report success rate $\eta_s$, violation rate $\eta_v$, and other rate $\eta_o$. The default zone size is 0.4, and the default number of zones per atomic proposition is 2. The evaluated specifications are the reach-avoid specifications with $n = 2$ shown in Table 4.

**Environment complexity.** We next investigate the impact of varying environment complexity performance. The results are shown in Figure 5. As the zone size increases or the number of zones associated with each atomic proposition increases, satisfying the safety constraints becomes more challenging. While all methods experience performance degradation, our method achieves the highest success rates and the lowest violation rate, which highlights its better ability to comply with safety constraints and generalize under environment variations.

**Unsatisfiable subgoals.** To evaluate the subgoal-switching mechanism, we use a satisfiable specification that contains an unsatisfiable subgoal: $\phi_1 : \neg\text{yellow U } ((\text{blue} \wedge \text{green}) \vee \text{magenta})$ in `ZoneEnv`. We randomly generate region layouts such that the green and blue regions (without overlaps) are positioned closer to the agent than the magenta region. In this setup, $\text{blue} \wedge \text{green}$ is usually selected as the first subgoal based on the learned value functions. As shown in Figure 6, the baselines struggle to satisfy the specification: the subgoal is unsatisfiable due to non-overlapping regions, and the agent gets stuck due to the lack of a mechanism to switch to an alternative. In contrast, our method can escape this deadlock by selecting an alternative subgoal. The trajectories in Figure 11 in Appendix C further illustrate this behavior.

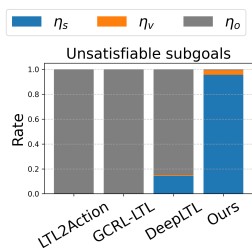

Figure 6: Evaluation results on tasks with unsatisfiable subgoals.

### 5.3 Ablation study: what are the individual impacts of the proposed observation reduction and safe RL approach (Q3)?

We consider a `ZoneEnv` environment with one region per color and reach-avoid specifications with a sequence length of $n = 2$. The environment layout is randomized in a controlled manner such that the region to be avoided is placed between the agent and the region to be reached according to an LTL specification. Under this setting, the agent must first circumvent the avoid region before reaching the goal region. For example, consider $\neg\text{magenta U blue}$, the `magenta` area is placed between the agent and the `blue` area. We construct a variant of our method by removing the observation reduction module and instead encoding the subgoal using the bitvector encoding discussed in Section 4.1. We also include DeepLTL as a reference, as it achieves the best performance among the baselines. The results are shown in Figure 7. We can observe that integrating the state-wise constraints leads to improved safety performance compared to the baseline method, and further incorporating the subgoal-induced observation reduction technique improves performance by significantly reducing the input dimension of the policy.

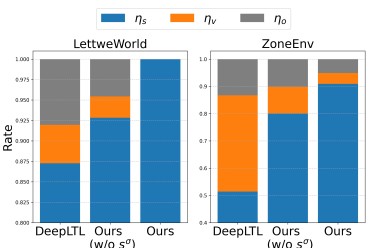

Figure 7: Average performance of success rate $\eta_s$, violation rate $\eta_v$, others rate $\eta_o$ for Ours(w/o $s^\sigma$), Ours and DeepLTL for reference.

## 6 Conclusion

This paper presents a novel method for learning RL policies that generalize, in a zero-shot manner, to arbitrary LTL specifications by performing subgoal decomposition through the corresponding Büchi automata and completing one subgoal at a time. Empirical results demonstrate that our method, despite being myopic, achieves substantially better generalization to complex and unseen specifications compared to existing baselines. We posit that the real complexity of LTL task generalization does not lie in the structure of the LTL formula or the corresponding automata, but rather in the combinatorial nature of the atomic propositions associated with the MDP states. The observation reduction technique that we introduce is a step towards addressing this complexity. Future work will explore alternative techniques that can exploit the Boolean structure within a subgoal, such as Boolean task algebra [37], with the goal of further improving the sample efficiency and generalization ability of our method.

## Acknowledgement

The authors thank the anonymous reviewers for their invaluable feedback and constructive suggestions. This work was supported in part by the U.S. National Science Foundation under grant CCF-2340776.

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

## Broader Impacts

Our work aims to improve the generalization of reinforcement learning to satisfy arbitrary LTL specifications. This framework enables flexible and expressive task definitions, but it also places responsibility on the user to ensure that the specifications are well designed. The behavior of the agent is tightly coupled with the given specification, so poorly constructed or harmful specifications may lead to unsafe or unintended behaviors. Thus, careful design and verification of specifications are essential when applying such methods in real-world settings.

## A Experimental Settings

### A.1 Environments

We use the `LetterWorld` [50] and `ZoneEnv` [50, 23] for evaluation. The `LetterWorld` is a $7 \times 7$ grid world that contains 12 letters corresponding to atomic propositions $AP = \{a, b, \ldots, l\}$. Each letter appears twice and is randomly placed on the grid. The observation consists of the entire grid in an egocentric view. The agent can move in four directions: up, down, left, and right. The grid wraps around, i.e., if the agent moves out of bounds, it reappears on the opposite side. The maximum episode length is $T = 75$. The `ZoneEnv` contains different colored regions corresponding to atomic propositions $AP = \{\text{blue}, \text{green}, \text{magenta}, \text{yellow}\}$ and walls acting as boundaries. Our implementation is adapted from [23]. We use the `point` robot that has a continuous action space for acceleration and steering. Observations consist of Lidar measurements of the colored regions and egocentric state information from other sensors. The maximum episode length is $T = 1000$. We further construct several variants of the `ZoneEnv` to evaluate the generalization performance of our method and baselines by modifying: (1) the region size, with $r = \{0.2, 0.3, 0.4(\text{default}), 0.5, 0.6\}$; (2) the number of regions per atomic proposition, with count $= \{1, 2(\text{default}), 3, 4, 5\}$; (3) and the number of atomic propositions, expanded to include new symbols $\{\text{red}, \text{cyan}, \text{orange}, \text{purple}, \text{teal}, \text{lime}\}$. Illustrations of the environments are provided in Figure 8 and trajectories under different LTL specifications are provided in Figure 12, 15, and 16.

We exclude the `FlatWorld`[23, 45] environment from our evaluation. It contains colored regions similar to `ZoneEnv`. However, the region layout is fixed, meaning that certain Boolean formulas over the atomic propositions cannot be evaluated since they are never true in this environment. Additionally, the observation space is limited to the agent's $(x, y)-$position. Due to this lack of variability and expressiveness, we exclude `FlatWorld` from our evaluation.

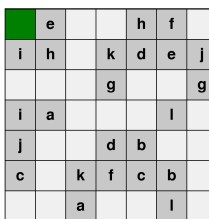 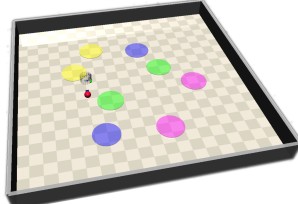

Figure 8: Visualisations of environments. Left: `LetterWorld`. Right: `ZoneEnv`

### A.2 LTL Specifications

We evaluate zero-shot generalization across a range of specifications, including complex and nested finite-horizon tasks (Table 1), complex infinite-horizon tasks (Table 1), varying specification complexity with different sequence lengths of reach-only and reach-avoid specifications (Table 4), varying environment complexity in reach-avoid tasks (Table 4), and different numbers of atopic propositions (Table 5). The specifications $\varphi_1 - \varphi_4$ and $\varphi_9 - \varphi_{12}$ are from [23]. To better evaluate each method's generalization ability and its handling of the trade-off between task progression and safety constraints, we construct more complex specifications, $\varphi_5-\varphi_8$ and $\varphi_{13}-\varphi_{16}$, which feature longer temporal sequences and stricter safety requirements, i.e., more atomic propositions appear under negation.

Table 4: Reach-only and reach-avoid specifications. Evaluation results are shown in Figure 4.

| | | |
|---|---|---|
| **Reach-only** | n=4 | F (yellow ∧ F (blue ∧ F (green ∧ F magenta))) |
| | | F (green ∧ F (magenta ∧ F (blue ∧ F yellow))) |
| | | F (blue ∧ F (yellow ∧ F (magenta ∧ F green))) |
| | n=6 | F (green ∧ F (blue ∧ F (magenta ∧ F (yellow ∧ F (blue ∧ F green))))) |
| | | F (yellow ∧ F (green ∧ F (magenta ∧ F (blue ∧ F (yellow ∧ F green))))) |
| | | F (blue ∧ F (magenta ∧ F (green ∧ F (yellow ∧ F (blue ∧ F magenta))))) |
| | n=8 | F (blue ∧ F (yellow ∧ F (green ∧ F (magenta ∧ F (blue ∧ F (green ∧ F (yellow ∧ F magenta))))))) |
| | | F (green ∧ F (blue ∧ F (yellow ∧ F (magenta ∧ F (green ∧ F (blue ∧ F (magenta ∧ F yellow))))))) |
| | | F (magenta ∧ F (yellow ∧ F (green ∧ F (blue ∧ F (magenta ∧ F (blue ∧ F (green ∧ F yellow))))))) |
| | n=10 | F (magenta ∧ F (yellow ∧ F (green ∧ F (blue ∧ F (magenta ∧ F (green ∧ F (yellow ∧ F (blue ∧ F (green ∧ F yellow))))))))) |
| | | F (blue ∧ F (green ∧ F (yellow ∧ F (magenta ∧ F (blue ∧ F (yellow ∧ F (magenta ∧ F (blue ∧ F yellow))))))))) |
| | | F (yellow ∧ F (blue ∧ F (magenta ∧ F (green ∧ F (yellow ∧ F (blue ∧ F (magenta ∧ F (green ∧ F (blue ∧ F magenta))))))))) |
| | n=12 | F (blue ∧ F (yellow ∧ F (magenta ∧ F (green ∧ F (blue ∧ F (yellow ∧ F (green ∧ F (magenta ∧ F (blue ∧ F (green ∧ F (yellow ∧ F magenta))))))))))) |
| | | F (magenta ∧ F (blue ∧ F (green ∧ F (yellow ∧ F (magenta ∧ F (blue ∧ F (yellow ∧ F (green ∧ F (magenta ∧ F (blue ∧ F (green ∧ F yellow))))))))))) |
| | | F (yellow ∧ F (magenta ∧ F (blue ∧ F (green ∧ F (yellow ∧ F (blue ∧ F (magenta ∧ F (green ∧ F (blue ∧ F (yellow ∧ F (magenta ∧ F green))))))))))) |
| **Reach-avoid** | n=2 | (¬blue ∧ ¬green) U yellow |
| | | (¬yellow ∧ ¬magenta) U green |
| | | (¬green ∧ ¬magenta) U blue |
| | n=4 | ¬(blue ∨ yellow) U (green ∧ (¬(blue ∨ magenta) U yellow)) |
| | | ¬(green ∨ yellow) U (magenta ∧ (¬(blue ∨ yellow) U green)) |
| | | ¬(yellow ∨ magenta) U (blue ∧ (¬(green ∨ magenta) U yellow)) |
| | n=6 | ¬(blue ∨ yellow) U (green ∧ (¬(blue ∨ magenta) U (yellow ∧ (¬(green ∨ blue) U magenta)))) |
| | | ¬(yellow ∨ blue) U (magenta ∧ (¬(green ∨ yellow) U (blue ∧ (¬(green ∨ magenta) U yellow)))) |
| | | ¬(green ∨ yellow) U (magenta ∧ (¬(green ∨ blue) U (yellow ∧ (¬(blue ∨ magenta) U green)))) |
| | n=8 | ¬(green ∨ blue) U (magenta ∧ (¬(yellow ∨ blue) U (green ∧ (¬(blue ∨ yellow) U (magenta ∧ (¬(yellow ∨ blue) U green)))))) |
| | | ¬(green ∨ yellow) U (magenta ∧ (¬(green ∨ blue) U (yellow ∧ (¬(green ∨ magenta) U (blue ∧ (¬(magenta ∨ yellow) U green)))))) |
| | | ¬(magenta ∨ blue) U (yellow ∧ (¬(green ∨ magenta) U (blue ∧ (¬(magenta ∨ green) U (yellow ∧ (¬(green ∨ blue) U magenta)))))) |
| | n=10 | ¬(magenta ∨ blue) U (green ∧ (¬(yellow ∨ blue) U (magenta ∧ (¬(yellow ∨ green) U (blue ∧ (¬(magenta ∨ green) U (yellow ∧ (¬(blue ∨ magenta) U green)))))))) |
| | | ¬(green ∨ blue) U (magenta ∧ (¬(green ∨ blue) U (yellow ∧ (¬(blue ∨ yellow) U (green ∧ (¬(magenta ∨ yellow) U (blue ∧ (¬(magenta ∨ green) U yellow)))))))) |
| | | ¬(yellow ∨ blue) U (green ∧ (¬(blue ∨ magenta) U (yellow ∧ (¬(magenta ∨ blue) U (green ∧ (¬(yellow ∨ blue) U (magenta ∧ (¬(green ∨ yellow) U blue)))))))) |

Table 5: Reach-avoid specifications with more APs. Evaluation results are shown in Table 6.

| | | |
|---|---|---|
| **ZoneEnv** | \|AP\| = 4 | ¬(blue ∨ yellow) U (green ∧ (¬(blue ∨ magenta) U (yellow ∧ (¬(green ∨ blue) U magenta)))) |
| | | ¬(yellow ∨ blue) U (magenta ∧ (¬(green ∨ yellow) U (blue ∧ (¬(green ∨ magenta) U yellow)))) |
| | | ¬(green ∨ yellow) U (magenta ∧ (¬(green ∨ blue) U (yellow ∧ (¬(blue ∨ magenta) U green)))) |
| | \|AP\| = 6 | ¬(red ∨ cyan) U (blue ∧ (¬(yellow ∨ green) U (magenta ∧ (¬(cyan ∨ yellow) U green)))) |
| | | ¬(green ∨ red) U (cyan ∧ (¬(blue ∨ magenta) U (yellow ∧ (¬(cyan ∨ red) U blue)))) |
| | | ¬(yellow ∨ cyan) U (red ∧ (¬(green ∨ blue) U (magenta ∧ (¬(red ∨ yellow) U green)))) |
| | \|AP\| = 8 | ¬(orange ∨ red) U (cyan ∧ (¬(blue ∨ green) U (yellow ∧ (¬(purple ∨ blue) U magenta)))) |
| | | ¬(purple ∨ cyan) U (orange ∧ (¬(red ∨ yellow) U (green ∧ (¬(magenta ∨ red) U blue)))) |
| | | ¬(yellow ∨ purple) U (green ∧ (¬(cyan ∨ orange) U (blue ∧ (¬(magenta ∨ green) U orange)))) |
| | \|AP\| = 10 | ¬(teal ∨ magenta) U (orange ∧ (¬(lime ∨ blue) U (yellow ∧ (¬(green ∨ purple) U red)))) |
| | | ¬(cyan ∨ purple) U (magenta ∧ (¬(red ∨ teal) U (lime ∧ (¬(orange ∨ yellow) U green)))) |
| | | ¬(blue ∨ green) U (yellow ∧ (¬(teal ∨ lime) U (purple ∧ (¬(magenta ∨ red) U orange)))) |

## B    Implementation Details

We use the official codebases of the baselines: LTL2Action[1], GCRL-LTL[2], RAD-embeddings[3], and DeepLTL[4]. Our implementation is adapted from DeepLTL by incorporating a safety value function and a state-dependent Lagrangian multiplier and removing DeepLTL-specific modules. For DeepLTL, LTL2Action, and our method, we use a fully connected actor network with three hidden layers of sizes [64, 64, 64]. The value function network has two hidden layers of sizes [64, 64]. For our method, both the safety value function and the Lagrangian multiplier network share the same architecture as the value function. GCRL-LTL employs a larger actor-critic architecture with [512, 1024, 256] units due to its use of a graph neural network. For fair comparison, we use the Adam optimizer with a learning rate of $3e-4$ and train all methods for 15M environment interactions. The discount factor is set to $\gamma = 0.94$ for LetterWorld and $\gamma = 0.998$ for ZoneEnv. Additional hyperparameter details are provided in the code in the supplementary materials. We train all the methods using a 16-core AMD CPU and an NVIDIA GeForce RTX 4090 GPU, and it takes roughly 3 hours to train our model.

Our method is summarized in Algorithm 1. We begin by constructing a subgoal set that includes all feasible subgoals: we first enumerate each assignment $\alpha \in 2^{AP}$ as a candidate $\alpha^+$. For each such $\alpha^+$, we filter out the remaining assignments that conflict with it. We then enumerate all possible

---

[1] https://github.com/LTL2Action/LTL2Action
[2] https://github.com/RU-Automated-Reasoning-Group/GCRL-LTL
[3] https://github.com/RAD-Embeddings/neurips24
[4] https://github.com/mathiasj33/deep-ltl

combinations of the filtered assignments to form $A^-$, resulting in the full set of feasible reach-avoid subgoals $\xi = \{(\alpha^+, A^-)_i\}_{i=1}^M$, from which subgoals are sampled uniformly. The agent receives a reward $r = 1$ if the subgoal is satisfied and $r = 0$ otherwise. At each timestep, it receives a safety signal $h = -1$ if the subgoal is not violated, and $h = 1$ otherwise. The reward and safety functions are defined as $r = \mathbb{1}[L(s^\sigma) \in \alpha^+]$ and $h = 2 \cdot \mathbb{1}[L(s^\sigma) \in A^-] - 1$, respectively. If the current subgoal is satisfied, we sample a new subgoal whose avoid set does not contain the current state, i.e., $L(s_t) \notin A^-$, and whose reach set also excludes the current state, i.e., $L(s_t) \notin \alpha^+$, to prevent the new subgoal from being immediately violated or trivially satisfied. If a subgoal is violated, the episode terminates. We use Generalized Advantage Estimation (GAE) [42] to estimate value functions:

$$A_r^\pi = \sum_{l=0}^{T-t-1}(\gamma\lambda_{\text{GAE}})^l\delta_{r,t+l} \quad \delta_{r,t} = r_t + \gamma V(s_{t+1}^\sigma) - V(s_t^\sigma) \tag{4}$$

$$A_h^\pi = \sum_{l=0}^{T-t-1}(\gamma\lambda_{\text{GAE}})^l\delta_{h,t+l} \quad \delta_{h,t} = (1-\gamma)h_t + \max(h_t, V(s_{t+1}^\sigma)) \tag{5}$$

where $\lambda_{\text{GAE}} = 0.95$ is the discount factor of GAE. At test time, given a target LTL specification, we convert it into a Büchi automaton and apply the DFS to enumerate all possible paths to accepting states. For each path, we evaluate the current subgoal using the learned value functions and select the optimal subgoal to execute. This process is applied iteratively until an accepting state is reached.

---

**Algorithm 1** GenZ-LTL

---

**Require:** Initial policy $\pi$, value and safety value function $V_r$, $V_h$, state-dependent Lagrangian multiplier $\lambda$, subgoal set $\xi$, maximum training iterations $K$, interactions per iteration $N$
1: **for** $k = 0, 1, 2, \ldots, K$ **do**
2:     **for** $n = 0, 1, 2, \ldots, N$ **do**
3:         **if** the current subgoal is not specified **or** the current subgoal is satisfied/violated **then**
4:             Sample a subgoal $\sigma \sim \text{Unif}(\xi)$
5:         **end if**
6:         Collect trajectory $\tau = \{s_t^\sigma, a_t, r_t, h_t\}$
7:     **end for**
8:     Compute reward-to-go $\hat{R}_t \doteq \sum_{i=t}^T \gamma^i r_i$ and cost-to-go $\hat{H}_t \doteq \max_t h_t$ for each $\tau$
9:     Compute advantage functions $A_r^\pi, A_h^\pi$, based on $V_r$ and $V_h$ using (4) and (5).
10:    Compute the Lagrangian multiplier $\lambda$.
11:    Fit value function, safety value function by regression on mean-square error.
12:    Update the policy parameters by maximizing (3).
13:    Update the multiplier parameters by minimizing (3).
14: **end for**

---

### B.1 Subgoal-induced observation reduction.

For Lidar measurements in the `ZoneEnv`, $f(\cdot)$ is implemented as the element-wise $\min$ to obtain the closest distance along each Lidar beam, such that the resulting $s^\sigma$ For grid maps in `LetterWorld`, we assign distinct values to each cell associated with $\sigma$ at location $(i, j)$: $f(s_{i,j}) = v_{\text{reach}}$ if $L(s_{i,j}) = \alpha^+$; $v_{\text{avoid}}$ if $L(s_{i,j}) \in A^-$; $v_{\text{neutral}}$ otherwise. This reduction allows the agent to focus on "reach" and "avoid" rather than specific set of atomic propositions and eliminates the need of encoding subgoals, enabling efficient policy learning and generalization to new atomic propositions that satisfies the conditioned discussed in Section 4.2 as shown later in Appendix C. If multiple observation functions are available, each capable of observing an assignment $\alpha \in 2^{AP}$, we can first fuse the observations within each observation type based on the current subgoal $\sigma$ and then concatenate the results to form the final state $s^\sigma$. In cases where each atomic proposition is tied to a distinct observation type, our method can be applied; however, learning policies in such settings is inherently intractable due to the exponential number of subgoal-state combinations.

The actual implementation of the observation reduction may appear straightforward, as the fusion operator is environment/sensor-specific. However, the idea behind this technique is broadly applicable. When solving the reach-avoid problem, what truly matters is whether a state satisfies the reach or avoid condition under the current subgoal, rather than its specific label $L(s)$ as we mentioned in Section 4.2. Moreover, our method can be extended to more realistic settings with some additional effort, but without requiring fundamental changes to the core approach. For example, in autonomous driving scenarios with multiple sensors such as cameras and LiDAR, observation reduction can be enabled through perception modules. Semantic segmentation can be applied to both image and point

cloud data to extract relevant features based on the current reach-avoid subgoals. For instance, if the avoid subgoal involves avoiding other cars and pedestrians, and the reach subgoal involves staying within the current lane, then perception can be used to segment the scene accordingly. This allows the agent to focus on task-relevant information and use processed observations, rather than raw sensory data, to learn value functions and policies effectively.

---

**Algorithm 2** Timeout-based subgoal-switching mechanism

---

**Require:** Trained agent, Büchi automaton $\mathcal{B}_\varphi := (\mathcal{Q}, \Sigma, \delta, \mathcal{F}, q_0)$, timeout value $t_{\max}$
1:  $\mathcal{Q}_c \leftarrow \{q_0\}$                                                           ▷ Current set of states.
2:  $\mathcal{U} \leftarrow \emptyset$                                                   ▷ Set of unsatisfiable transitions.
3:  **while** FindSubgoals($\mathcal{B}_\varphi, \mathcal{Q}_c, \mathcal{U}) \neq \emptyset$ **do**
4:     Select subgoal $\sigma = (\alpha^+, A^-)$ from FindSubgoals($\mathcal{B}_\varphi, \mathcal{Q}_c, \mathcal{U}$) with maximum value.
5:     **for** $t = 1, 2, \ldots, t_{\max}$ **do**
6:         Get an action from the agent, apply it, and observe $\alpha \in \Sigma$.
7:         $\mathcal{Q}_o \leftarrow \mathcal{Q}_c$                                ▷ Remember the old states.
8:         $\mathcal{Q}_c \leftarrow \{q' \mid \exists q \in \mathcal{Q}_o, (q, \alpha, q') \in \delta\}$         ▷ Update the current set of states.
9:         **if** $\mathcal{Q}_c \neq \mathcal{Q}_o$ **then**
10:           **break**               ▷ A new state set is reached; the subgoal should be updated.
11:         **else if** $t = t_{\max}$ **then**
12:           $\mathcal{U} \leftarrow \mathcal{U} \cup \{(q, \alpha^+) \mid q \in (\mathcal{Q}_c - \mathcal{F})\}$     ▷ If $q$ is not accepting, mark this subgoal
                                                                     as unsatisfiable.
13:         **end if**
14:     **end for**
15: **end while**
16: All subgoals are marked as unsatisfiable. Terminate.

---

### B.2 Timeout & Subgoal-switching Mechanism

At the test time, given a target LTL specification, we convert it into a Büchi automaton and apply a depth-first search (DFS) to enumerate all possible paths to accepting states. However, not all of these paths, or in some cases, any, are feasible in the physical environment, as some may include subgoals that are unsatisfiable. Although the labeling function is known, meaning we can determine which propositions are true for any given state, it does not necessarily imply that we can exhaustively iterate over the entire state space to identify unsatisfiable assignments.

For example, consider the specification ¬yellow U ((blue ∧ green) ∨ magenta). To satisfy this formula, the agent must eventually reach either a magenta region or a region where blue and green overlap. However, the latter becomes unsatisfiable if the blue and green regions are disjoint. In such cases, the agent must be able to adapt by pursuing an alternative subgoal, a capability that is essential for goal-conditioned policies, including our approach as well as existing baselines such as DeepLTL and GCRL-LTL. However, these baselines do not address this issue.

To address this, we implement a timeout mechanism that allows the agent to revisit the automaton and search for alternative subgoals when the current one cannot be satisfied. We determine subgoal satisfiability by enforcing a timeout threshold $(1 + \epsilon_{\text{scale}}) \cdot \mu_{\text{subgoal}}$ as discussed in Section 4.1: if the agent fails to achieve this subgoal within that timestep, it is considered unsatisfiable, and the agent switches its subgoal. If no feasible subgoals remain, the episode terminates.

This timeout & subgoal-switching mechanism is explained in Algorithm 2, which outlines the general mode of operation during the test time. We use a set $\mathcal{U} : \mathcal{Q} \times \Sigma$ to store the transitions that are marked as unsatisfiable after a timeout. These unsatisfiable transitions are ignored by the FindSubgoals function. Apart from this, the only difference between FindSubgoals and the DFS algorithm used in DeepLTL is that we separate the subgoals after identifying the paths with accepting cycles.

## C   Additional Experiments

**Training curves.** We first evaluate simple reach-only and reach-avoid specifications with sequence length of $n = 3$ every specific training interval, and the success and violation rates are shown in Figure 9. We can observe that our method outperforms the baselines in terms of achieving a lower

violation rate (close to zero constraint violation) while maintaining a high success rate. The violation rates of the baselines also decrease over training because task progression and safety constraints are jointly encoded in the LTL specification, so optimizing for satisfaction of the specification implicitly improves safety performance. However, due to the inherent trade-off between these requirements, the baselines struggle to further reduce violation rates. In contrast, by formulating the problem using safe RL framework with reachability constraints, our method achieves stricter constraint satisfaction, thereby improving the overall probability of satisfying the full LTL specifications.

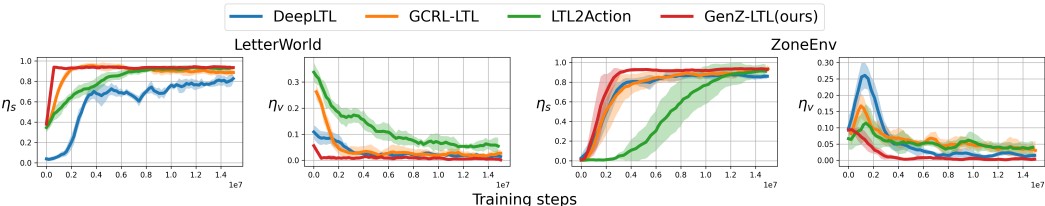

Figure 9: Evaluation curves on simple reach-only and reach-avoid specifications with $n = 3$. Each value is averaged over 5 seeds with 50 trajectories with randomly sampled tasks.

**Partial observability.** To satisfy arbitrary LTL specifications, we leverage the structure of the Büchi automaton to decompose each formula into a sequence of subgoals and handle one subgoal at a time, progressing through the sequence incrementally. While our method could be extended to condition on the full subgoal sequence, we argue that this introduces significant drawbacks that outweigh the potential benefits. Among the baselines, DeepLTL conditions its policy on the full subgoal sequence to achieve non-myopic behavior. Intuitively, this "look-ahead" capability may help the agent satisfy the specification in fewer steps. However, subgoal sequences come with several disadvantages. First, policies trained on full sequences are more likely to encounter OOD issues at test time, as the sequence length can vary across specifications. Although one might limit the policy to observe at most $k$ subgoals, where $k$ is the maximum length seen during training, choosing $k$ is nontrivial. A large $k$ can lead to exponential sample complexity due to the combinatorial explosion of subgoal sequences, e.g., each subgoal is drawn from $2^{AP}$, while a small $k$ limits the look-ahead benefit. Moreover, in practice, the agent may not perceive all relevant subgoals at once, rendering the subgoal sequence ineffective. To illustrate this, we conduct experiments where the agent's observability is restricted to half of the environment. We train both DeepLTL and our method in `LetterWorld` and `ZoneEnv`, and then evaluate them using specifications $\varphi_1$–$\varphi_3$ and reach-avoid tasks with $n = 4$. We report the average success rate $\eta_s$ and violation rate $\eta_v$. As shown in the left and middle plots of Figure 10, our single-subgoal-at-a-time method outperforms DeepLTL in terms of both the success and violation rates, despite the latter using full subgoal sequences.

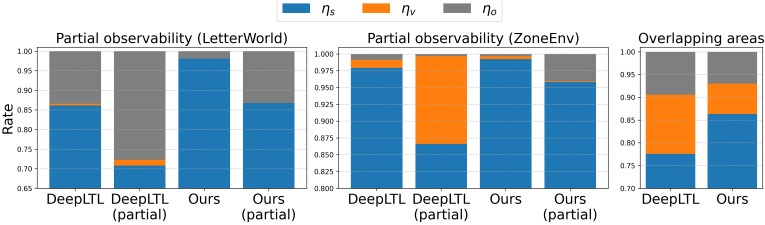

Figure 10: Evaluation results of DeepLTL and our method in two settings: (1) partial observability (left two plots) and (2) overlapping areas (the rightmost plot).

**Overlapping areas.** We construct a variant of `ZoneEnv` to introduce a more challenging setting where multiple atomic propositions can be true simultaneously, i.e., different colored regions may overlap in arbitrary configurations. Note that `FlatWorld` is an overly simplified case where the layout is fixed. In `ZoneEnv`, overlapping areas can be handled in two ways: (1) by adding separate observations for each assignment involving multiple true propositions, or (2) by using observations only for individual atomic propositions (default setting for `ZoneEnv`) and learning policies through interaction with the environment, during which the agent implicitly discovers the relationships among propositions and observations. The first approach does not offer meaningful advantages over the second, as our observation reduction technique can extract the relevant parts of the observation

for any target assignment and fuse them as needed. Thus, we adopt the second approach, which presents a more challenging generalization problem. To accommodate this scenario, we modify DeepLTL's curriculum training to allow multiple atomic propositions to be true at the same time. The positions of both the agent and the regions are randomly sampled. We evaluate both methods using reach-avoid specifications with $n = 2$: $\phi_2 : \neg(\text{magenta} \lor \text{yellow}) \text{ U } (\text{blue} \land \text{green})$, $\phi_3 : \neg(\text{blue} \lor \text{magenta}) \text{ U } (\text{yellow} \land \text{green})$, $\phi_4 : \neg(\text{yellow} \lor \text{blue}) \text{ U } (\text{green} \land \text{magenta})$, and $\phi_5 : \neg(\text{green} \lor \text{magenta}) \text{ U } (\text{yellow} \land \text{blue})$. Trajectory illustrations are shown in Figure 11. We report the average rates and the results in the rightmost plot in Figure 10 show that our method achieves a higher success rate and a lower violation rate. Our method can better satisfy safety constraints due to the integration of reachability constraints.

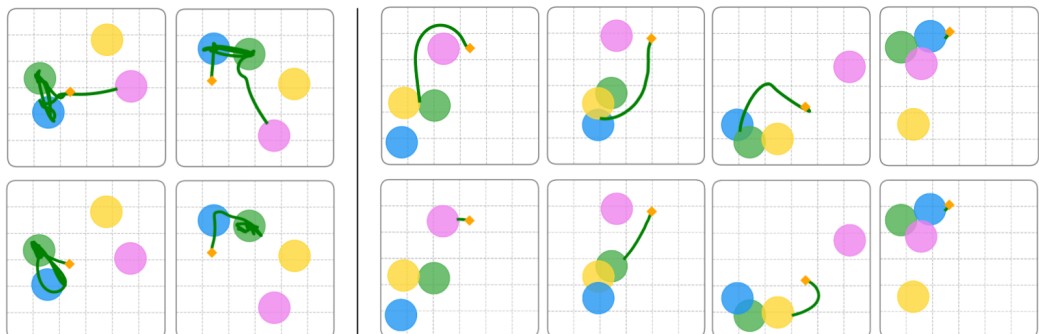

Figure 11: Illustration of trajectories in `ZoneEnv`: (1) unsatisfiable subgoals (left two columns, $\phi_1$) and (2) overlapping areas (right four columns, $\phi_2$–$\phi_5$). The top row shows trajectories generated by our method, while the bottom row shows those from DeepLTL.

**Number of atomic propositions.** We also evaluate performance as the number of atomic propositions increases. In this experiment, we emphasize that the observations corresponding to the additional atomic propositions satisfy the conditions outlined in Section 4.2. It is important to note that when new atomic propositions are observed through modalities not encountered during training — such as those representing the agent's speed or heading — zero-shot adaptation remains an open challenge. The results are presented in Table 6, and example trajectories are shown in Figure 12. Among the evaluated methods, only our method can generalize in a zero-shot manner to new atomic propositions whose observations are produced by identical observation functions under output transformation, as the subgoal-induced observation reduction technique fuses these propositions into "reach" and "avoid", allowing the agent to make decisions without relying on their specific labels $L(s)$.

Table 6: Performance in `ZoneEnv` with increasing number of atomic propositions. We evaluate the reach-avoid specifications with sequence length of $n = 6$, using an increasing number of atomic propositions as listed in Table 5. We report success rate $\eta_s$, violation rate $\eta_v$, other rate $\eta_o$, and average steps to satisfy the specifications $\mu$.

| Metrics | \|AP\| = 4 | \|AP\| = 6 | \|AP\| = 8 | \|AP\| = 10 |
|---|---|---|---|---|
| $\eta_s \uparrow$ | $0.96_{\pm 0.02}$ | $0.94_{\pm 0.02}$ | $0.91_{\pm 0.03}$ | $0.93_{\pm 0.03}$ |
| $\eta_v \downarrow$ | $0.02_{\pm 0.01}$ | $0.02_{\pm 0.01}$ | $0.03_{\pm 0.01}$ | $0.02_{\pm 0.01}$ |
| $\eta_o \downarrow$ | $0.02_{\pm 0.02}$ | $0.04_{\pm 0.02}$ | $0.06_{\pm 0.03}$ | $0.05_{\pm 0.02}$ |
| $\mu \downarrow$ | $306.58_{\pm 16.88}$ | $314.88_{\pm 17.02}$ | $313.10_{\pm 10.61}$ | $323.30_{\pm 12.57}$ |

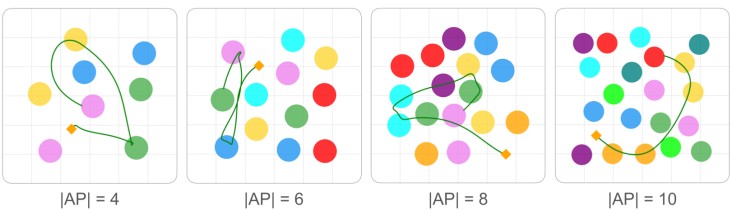

|AP| = 4      |AP| = 6      |AP| = 8      |AP| = 10

Figure 12: Illustration of `ZoneEnv` with an increasing number of atomic propositions. The trajectories shown correspond to the first specification for $|\text{AP}| = 4, 6, 8, 10$ in Table 5.

**Deterministic v.s. stochastic policies.** As discussed in Section 5, we follow the standard practice of using a deterministic policy during evaluation. We also note that DeepLTL is evaluated using a stochastic policy in `LetterWorld` in its original paper. To provide a comprehensive comparison, we report results for both stochastic and deterministic policies in `LetterWorld`, as shown in Figure 13. We observe that DeepLTL exhibits inconsistent behavior: while the success rate improves under a stochastic policy, the violation rate also increases. This trade-off is unacceptable in safety-critical applications and indicates that DeepLTL struggles to maintain safety. In contrast, our method delivers consistent performance without compromising safety for task progression, since we explicitly model the violation of subgoals as a reachability constant in a safe RL formulation. In our view, stochastic policies are not suitable for evaluation. For reach-only specifications, they contain no safety constraints; a stochastic policy may satisfy them simply by randomly exploring the environment, thereby overstating performance. For reach-avoid specifications, stochasticity further introduces inconsistency: the mean action may be safe, while a sampled action may lead to a violation, or vice versa. This randomness complicates evaluation and prevents a fair comparison between methods.

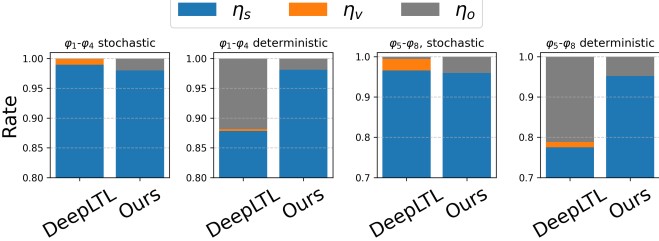

Figure 13: Evaluation results of DeepLTL and our method in `LetterWorld` using stochastic and deterministic policies. Each value is averaged over 5 seeds, with 100 trajectories per seed.

Table 7: Performance in `ZoneEnv` with Ant agent. We report the success rate $\eta_s$, violation rate $\eta_v$, and the average steps to satisfy the specifications $\mu$. Each value is averaged over 5 seeds, with 100 trajectories per seed.

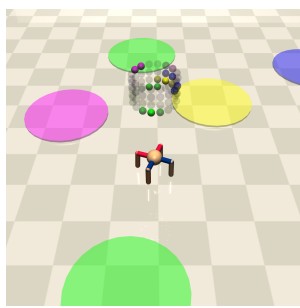

Figure 14: Visualization of the Ant agent in `ZoneEnv`.

| | | GenZ-LTL(ours) | | | DeepLTL | | |
| | | $\eta_s \uparrow$ | $\eta_v \downarrow$ | $\mu \downarrow$ | $\eta_s \uparrow$ | $\eta_v \downarrow$ | $\mu \downarrow$ |
|---|---|---|---|---|---|---|---|
| ZoneEnv | $\varphi_9$ | $0.97_{\pm0.02}$ | $0.02_{\pm0.01}$ | $305.32_{\pm46.50}$ | $0.00_{\pm0.01}$ | $0.23_{\pm0.08}$ | $865.50_{\pm72.83}$ |
| | $\varphi_{10}$ | $0.91_{\pm0.03}$ | $0.09_{\pm0.03}$ | $202.00_{\pm53.33}$ | $0.03_{\pm0.02}$ | $0.39_{\pm0.11}$ | $588.58_{\pm166.33}$ |
| | $\varphi_{11}$ | $0.95_{\pm0.02}$ | $0.04_{\pm0.02}$ | $182.88_{\pm38.66}$ | $0.07_{\pm0.02}$ | $0.27_{\pm0.05}$ | $604.87_{\pm28.22}$ |
| | $\varphi_{12}$ | $0.98_{\pm0.01}$ | $0.00_{\pm0.00}$ | $125.97_{\pm35.66}$ | $0.24_{\pm0.05}$ | $0.05_{\pm0.04}$ | $392.04_{\pm64.39}$ |
| | $\varphi_{9-12}$ | $0.95_{\pm0.04}$ | $0.04_{\pm0.04}$ | $204.04_{\pm77.69}$ | $0.09_{\pm0.10}$ | $0.23_{\pm0.14}$ | $568.15_{\pm172.57}$ |
| | $\varphi_{13}$ | $0.93_{\pm0.04}$ | $0.00_{\pm0.00}$ | $459.99_{\pm86.69}$ | $0.00_{\pm0.00}$ | $0.00_{\pm0.00}$ | - |
| | $\varphi_{14}$ | $0.95_{\pm0.02}$ | $0.00_{\pm0.00}$ | $434.41_{\pm76.87}$ | $0.00_{\pm0.00}$ | $0.00_{\pm0.00}$ | - |
| | $\varphi_{15}$ | $0.89_{\pm0.02}$ | $0.08_{\pm0.02}$ | $429.76_{\pm75.75}$ | $0.00_{\pm0.00}$ | $0.39_{\pm0.13}$ | - |
| | $\varphi_{16}$ | $0.95_{\pm0.03}$ | $0.00_{\pm0.00}$ | $449.89_{\pm83.26}$ | $0.00_{\pm0.00}$ | $0.00_{\pm0.00}$ | - |
| | $\varphi_{13-16}$ | $0.93_{\pm0.04}$ | $0.02_{\pm0.04}$ | $443.51_{\pm73.31}$ | $0.00_{\pm0.00}$ | $0.10_{\pm0.18}$ | - |

**Complex Dynamics.** To evaluate the performance of our method in environments with more complex dynamics, we conduct additional experiments using the Ant agent [25], consisting of a torso and four legs connected by hinge joints, in the Zone environment. The state space includes the agent's ego-state (40 dimensions, compared to 12 for the Point agent used in our main experiments) and LiDAR observations of the regions. A visualization of the Ant agent is shown in Figure 14. The action space has 8 dimensions, corresponding to the torques applied to the Ant's joints to coordinate leg movements (compared to 2 dimensions for the Point agent). We train and evaluate our method against DeepLTL, the strongest baseline in our main results. For a fair comparison, all training procedures, training parameters, and neural network sizes were kept the same (note that the last one in particular could result in some performance drop as the state dimension and action dimension are bigger for the ant agent). Both methods are trained until convergence. The evaluation results are shown in Table 7. We can observe that our method consistently outperforms the baseline across all metrics, demonstrating an even larger performance gain in environments with more complex agent dynamics. Note that for the Ant agent, in addition to the LTL specifications, there is a built-in safety constraint that the agent should not fall headfirst, enforced as a hard constraint with immediate episode termination. Although a negative reward is assigned in such cases, as is done when a specification is

violated for DeepLTL, it does not explicitly model safety constraints as our method does through HJ reachability, making it less effective in handling such cases. This impedes the learning of coordinated locomotion and then the satisfaction of LTL specifications, leading to degraded performance.

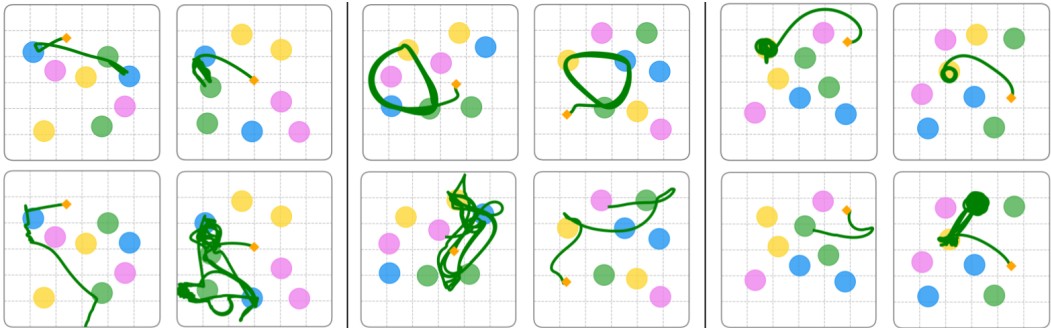

Figure 15: Illustration of infinite-horizon tasks in `ZoneEnv`. The trajectories correspond to specifications $\psi_1$–$\psi_3$, shown from left to right (two columns per specification). The top row shows trajectories generated by our method, while the bottom row shows those from DeepLTL.

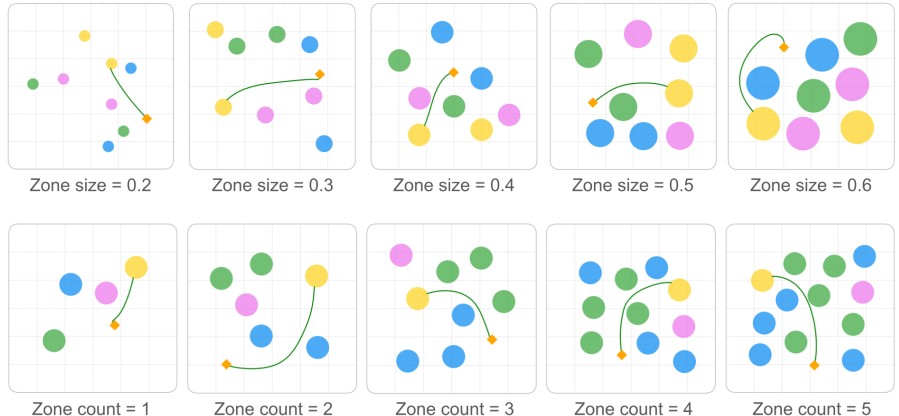

Figure 16: Illustration of `ZoneEnv` under increasing environment complexity, including different zone sizes and different numbers of zones per atomic proposition. The default zone size is 0.4, and the default number of zones per atomic proposition is 2. The trajectories are generated by our method and correspond to the evaluated specification $(\neg\text{blue} \wedge \neg\text{green}) \, \mathsf{U} \, \text{yellow}$.

## D  Visualizations

We provide visualizations for the experiments in Section 5.1 and Section 5.2, including infinite-horizon tasks (Figure 15), increasing environment complexity (Figure 16) and increasing numbers of atomic propositions (Figure 12). For infinite-horizon tasks, our method exhibits a clear recurring pattern that visits the specified regions in the correct order, while the baseline struggles to do so and suffers from safety constraint violations. For different environment complexities, we vary the zone size and zone count per atomic proposition. Importantly, we increase the zone count only for propositions the agent must avoid, since doing so for propositions the agent need to satisfy would simplify the task. Larger zones or more zones per proposition force the agent to navigate around constrained regions to reach the target, and the resulting trajectories demonstrate that our method satisfies the safety constraints. For experiments with an increasing number of atomic propositions, we incrementally introduce new propositions, up to 10 in total, each corresponding to a new colored region with associated lidar observations. This makes zero-shot generalization challenging for baseline methods, as they explicitly encode atomic propositions and cannot generalize to unseen ones without retraining. Even substituting one proposition for another, for example, replacing blue with red, causes these baselines to fail, due to their dependence on fixed encodings. In contrast, our method uses subgoal-induced observation reduction, which removes this dependency when atomic

propositions share the same property. This allows our method to focus on the semantics of the propositions, i.e., "reach" or "avoid", rather than the specific proposition symbols, enabling zero-shot generalization to unseen atomic propositions.

