# OpenReview forum: "One Subgoal at a Time: Zero-Shot Generalization to Arbitrary Linear Temporal Logic Requirements in Multi-Task Reinforcement Learning"
_NeurIPS.cc/2025/Conference — NeurIPS 2025 poster_

### Official Review · Reviewer_bwRF · 2025-06-21

**Clarity:** 4
**Significance:** 3
**Originality:** 4
**Rating:** 5
**Confidence:** 2

**Summary:**

This paper introduces GenZ-LTL, a method enabling RL agents to achieve zero-shot generalization for tasks specified in Linear Temporal Logic formulas. The algorithm decomposes LTL requirements into a series of subgoals derived from Buchi automata, which are then solved sequentially using safe RL. The authors confirm that GenZ-LTL works well in simulated navigation environments.

**Questions:**

1. Suggestion: a more in-depth explanation of the environments, starting in line 250, would be helpful. What is the goal of the agent in these environments? Figure 7 in the Appendix was helpful.
2. Line 54 oversells slightly, in that the environments are focused on navigation.
3. Line 239: Did you ever observe the greedy assumption to be an issue? I suppose if the agent were in an environment where it had to actively plan, then this assumption might be?
4. Line 180: Moving some detail up from appendix or having an algorithm block in the main text might be helpful for understanding here.

Thank you for your time!

**Ethical Concerns:**

["NO or VERY MINOR ethics concerns only"]

**Final Justification:**

The authors adequately address the limitations I brought up, centering around the toyness of the evaluation and how the method might be applied in real life. Reviewer NmCA brought up concerns about novelty, which I am unqualified to decide, since I am not familiar with the papers cited by NmCA. Accordingly, I have maintained my score and decreased my confidence from 3 to 2.

**Limitations:**

I think the paper would benefit from an explicit limitations section. Talking points that comes to mind:
1. The two environments tested both are fairly programmatic, and thus map cleanly onto the LTL specification. How do the authors envision this work generalizing to more realistic settings?
2. Can one have a perfect LTL specification? We assume here that the agent receives an LTL formula, if I understand correctly, but achieving this formula or description of the environment in the first place seems like a challenge.
3. How does this method interact with soft constraints or preferences? Do these come up in the intended use case, or can we assume clean alignment with LTL?

**Quality:**

4

**Strengths And Weaknesses:**

## Strengths
- The paper's core idea of solving LTL tasks from a goal-conditioned RL perspective is novel, and the results align with the expected benefits of goal-conditioned RL: better zero-shot and OOD generalization.
- Treating avoid subgoals as hard constraints is also technically impressive.
- Clear and high-quality writing.

## Weaknesses
The primary weakness I find is that the experimental validation, while extensive, is limited to programmatic navigation-style environments. These environments are particularly favorable for LTL, because tasks map cleanly onto rule-based constraints (go here, don't go here). However, one cannot do everything in one paper, so I simply recommend the authors acknowledge and address this in an explicit limitations section, separate from conclusion or broader impacts (more below).

---

> ### Author Rebuttal · Authors · 2025-07-30
>
> We thank the reviewer for their thoughtful comments and constructive feedback. We address each weakness, question, and limitation below.
>
> ## Weaknesess
> > 1. The primary weakness I find is that the experimental validation, while extensive, is limited to programmatic navigation-style environments. These environments are particularly favorable for LTL, because tasks map cleanly onto rule-based constraints (go here, don't go here). However, one cannot do everything in one paper, so I simply recommend the authors acknowledge and address this in an explicit limitations section, separate from conclusion or broader impacts (more below).
>
> We consider these navigation-style environments as they are also used as benchmarks in prior works [23, 29, 50], thus allowing us to conduct fair comparisons with state-of-the-art methods. We agree with the reviewer that more diverse experimental domains would further strengthen the evaluation. At present, suitable benchmarks for more complex LTL-guided tasks are limited, but we plan to evaluate our method in broader settings as such environments become available (we also have ongoing efforts focusing on creating additional benchmarks). We will explicitly note this in a dedicated limitations section in the next version of the paper.
>
> ## Questions
> > 1. Suggestion: a more in-depth explanation of the environments, starting in line 250, would be helpful. What is the goal of the agent in these environments? Figure 7 in the Appendix was helpful.
>
> We thank the reviewer for the suggestion. Due to page limit, we provide the environment details in Appendix A. In the revision, we will add a brief explanation in the main text to clarify the agent's goal in these environments, such as "the agent must reach certain regions and avoid others in some order based on the given LTL specification."
>
> > 2. Line 54 oversells slightly, in that the environments are focused on navigation.
>
> We will revise it as "we conduct comprehensive experiments across a set of navigation-style environments and diverse task specifications, demonstrating that our method consistently outperforms state-of-the-art baselines in zero-shot generalization to arbitrary LTL specifications."
>
> > 3. Line 239: Did you ever observe the greedy assumption to be an issue? I suppose if the agent were in an environment where it had to actively plan, then this assumption might be?
>
> First, we would like to emphasize that our subgoal-switching mechanism serves as a form of active planning, which is absent in baselines such as [23, 39, 50]. As demonstrated in the evaluation in Section 5.2, these baselines lack the ability to escape from deadlocks caused by unsatisfiable subgoals. Second, we highlight that the presence of unsatisfiable subgoals adds significant complexity to the problem. Consider the specification $(\mathbf{F}(a \land \mathbf{F} b)) \lor (\neg a \mathbf{U} c)$: the agent can either visit $a$ followed by $b$, or avoid $a$ until reaching $c$. Our method may choose the first strategy, visiting $a$ and then attempting $b$. However, if $b$ turns out to be unsatisfiable, the trajectory fails to satisfy the specification since visiting $a$ is irreversible, thus invalidating the second option. This issue is not unique to our method, it also affects baselines with global planning capabilities, such as DeepLTL, because the satisfiability of subgoals cannot generally be determined a priori as we discuss in lines 237-242.
>
> > 4. Line 180: Moving some detail up from appendix or having an algorithm block in the main text might be helpful for understanding here.
>
> We will move some details from the appendix to the main text, specifically, Appendix B.1, which explains how observation reduction is implemented under different observation settings, and place it after Section 4.2 to improve clarity and understanding of our method.
>
> ## Limitations
> > 1. The two environments tested both are fairly programmatic, and thus map cleanly onto the LTL specification. How do the authors envision this work generalizing to more realistic settings?
>
> We believe that our method can be extended to more realistic settings with some additional effort, but without requiring fundamental changes to the core approach. For example, in autonomous driving scenarios with multiple sensors such as cameras and LiDAR, observation reduction can be enabled through perception modules. Semantic segmentation can be applied to both image and point cloud data to extract relevant features based on the current reach-avoid subgoals. For instance, if the avoid subgoal involves avoiding other cars and pedestrians, and the reach subgoal involves staying within the current lane, then perception can be used to segment the scene accordingly. This allows the agent to focus on task-relevant information and use processed observations, rather than raw sensory data, to learn value functions and policies effectively.
>
> > 2. Can one have a perfect LTL specification? We assume here that the agent receives an LTL formula, if I understand correctly, but achieving this formula or description of the environment in the first place seems like a challenge.
>
> This is a very interesting and relevant question. Our work and also prior works on LTL task generalization consider the setting where a formal LTL specification is given as input. In practice, coming up with accurate, formal LTL specifications for complex task requirements can indeed be a challenge. There are many ongoing efforts in the formal methods community to address this challenge, such as automatically converting natural language specifications to formal specifications, designing domain-specific languages to facilitate such specifications, and mining formal specifications from system executions or user demonstrations. These efforts are orthogonal and complementary to LTL task generalization, which is the problem we focus on in this paper.
>
> > 3. How does this method interact with soft constraints or preferences? Do these come up in the intended use case, or can we assume clean alignment with LTL?
>
> In this paper, we consider problems that require clean alignment with the given LTL specification, i.e., either the LTL specification is satisfied or not satisfied. The semantics of LTL do not support soft constraints, but there are extensions such as Signal Temporal Logic (STL) that incorporate quantitative semantics and enable reasoning about robust satisfaction. There are some works on training RL policies to be aligned with different robustness values of a given STL specification [1, 2, 3]. However, they do not consider generalization to unseen specifications, and STL does not naturally admit an automaton-based representation, which is used for subgoal decomposition in our setting. It would be interesting future work to consider soft constraints or even preferences, but they do not reflect the intended use cases in this paper.
>
> [1] Leung, Karen, Nikos Aréchiga, and Marco Pavone. "Backpropagation through Signal Temporal Logic Specifications: Infusing Logical Structure into Gradient-based Methods." The International Journal of Robotics Research 2023.
> [2] Guo, Zijian, Weichao Zhou, and Wenchao Li. "Temporal Logic Specification-Conditioned Decision Transformer for Offline Safe Reinforcement Learning." International Conference on Machine Learning. PMLR, 2024.
> [3] Meng, Yue, and Chuchu Fan. "Signal Temporal Logic Neural Predictive Control." IEEE Robotics and Automation Letters 2023.

---

### Official Review · Reviewer_VrZq · 2025-06-26

**Clarity:** 3
**Significance:** 3
**Originality:** 3
**Rating:** 5
**Confidence:** 4

**Summary:**

The paper introduces a multi-task RL framework for tasks given as LTL specifications, combining goal-conditioned RL and safe RL. Specifically, for a given LTL task, they perform a DFS search to identify accepting cycles on the induced Buchi automaton and extract reach and avoid subgoals; then, if applicable, apply their proposed subgoal-induced observation reduction method to reduce the state size; and finally, they sequentially (and myopically) pass individual reach-avoid problems to a goal conditioned policy. They incorporate Hamilton-Jacobi reachability to their goal-conditioned policy by simultaneously learning a value function estimating the feasible set and using this value function with a state-dependent Lagrangian multiplier to select the subgoal that achieves the best trade-off between the task progression and the safety constraint. To account for the fact that some subgoals might be unsatisfiable, they implement a subgoal switching mechanism that is triggered if the current subgoal is not reached within a timeout threshold; if the current subgoal times out, then they switch to the next best subgoal, selected based on the value functions. During these attempts, they make sure that the current reach-avoid problem does not violate another reach-avoid problem by respecting all avoid subgoals of the current automaton state. An experimental evaluation of the method is presented, comparing the proposed framework against a selected set of previous work and showing its efficacy.

**Questions:**

1. Have you tried comparing against [54]? I think this empirical comparison is needed, as [54] is a newer technique that outperforms [50]. Also, do you agree with my comment that LTL formula embeddings [50] and automata embeddings [54] can be incorporated into your technique to overcome the myopia issue? If so, I think the paper can benefit from a discussion highlighting how these different approaches are orthogonal and can be incorporated to get the best of both worlds, i.e., overcome myopia via embeddings and the benefits of your safe RL formulation.
2. In Figure 6, what were the network sizes and number of training steps for "Ours (w/o s^\sigma)" and "Ours"? Were they the same? If so, then do you think that given more compute, "Ours (w/o s^\sigma)" could perform as well as "Ours"?
3. Does the benefit of the state reduction technique stem from the compression it provides through exploiting the symmetries in the observation? Is it possible to get the same performance with more compute when this state reduction technique is not used? I think a discussion or an experiment on this matter might be helpful to further highlight what this state reduction technique provides. Especially given the fact that it is an environment-specific approach that cannot be used in any environment, it would be good to have a further discussion of how to make the overall method work as well in cases where this state reduction technique cannot be used.

**Ethical Concerns:**

["NO or VERY MINOR ethics concerns only"]

**Final Justification:**

The authors have adequately addressed my concerns in their rebuttal. I am maintaining my rating (5) and think that this is a good paper worthy of acceptance.

**Limitations:**

yes

**Paper Formatting Concerns:**

No major formatting issues.

**Quality:**

3

**Strengths And Weaknesses:**

The paper combines the existing approaches for multi-task RL with LTL objectives and Hamilton-Jacobi-based safe RL in an intuitive and clear manner. Their extensive experimental evaluation demonstrates the benefits of their proposed approach compared to the previous work. Their subgoal-induced observation reduction technique is a novel contribution that stems from a simple observation about the local reach-avoid nature of LTL specifications; however, I found the explanation given in Sec 4.2 a bit too complicated. I wonder if that explanation can be simplified, or the authors might even consider adding a figure to assist the reader with the intuition that they are trying to communicate. Also, an experimental comparison with [54] is needed, as that work outperforms [50]. Furthermore, as also mentioned in the paper, the proposed approach is myopic since the goal-conditioned policy doesn't know what the future subgoals are. So, the authors might consider including an experimental evaluation (at least in the appendix) tailored to show this limitation to demonstrate how the proposed approach behaves compared to [50, 54] and DeepLTL in terms of its myopia. That being said, I think the approach presented in this paper is orthogonal to techniques given in [50, 54], in the sense that one can also include LTL formula embeddings [50] or automaton embeddings [54] in the product state and overcome myopia while still leveraging the benefits of the proposed framework. Overall, this is a well-written paper with good experimental evaluation, and such additional experiments and/or further discussion on these issues could make it a stronger contribution and help better position it in the literature.

---

> ### Author Rebuttal · Authors · 2025-07-30
>
> We are grateful for the thoughtful comments provided by the reviewer. In the following, we address the weaknesses and questions in detail.
>
> ## Weaknesses and Questions
> > 1. Have you tried comparing against [54]? I think this empirical comparison is needed, as [54] is a newer technique that outperforms [50]. Also, do you agree with my comment that LTL formula embeddings [50] and automata embeddings [54] can be incorporated into your technique to overcome the myopia issue? If so, I think the paper can benefit from a discussion highlighting how these different approaches are orthogonal and can be incorporated to get the best of both worlds, i.e., overcome myopia via embeddings and the benefits of your safe RL formulation.
>
> The evaluation results of [54] and our method (as in Table 1) are shown in the table below. We can see that our method achieves higher success rates in all instances. In addition, our method needs fewer steps to satisfy the specifications in all but two cases ($\varphi_{11}$ and $\varphi_{12}$), demonstrating that solving one subgoal at a time is effective also from a performance perspective compared with the state-of-the-art automata embedding-based approach. Moreover, achieving globally optimal behavior requires full observability of the environment states in general, a condition that is often unmet in practice. We elaborate on the performance difference below.
>
> [54] pre-trains the automaton embeddings by sampling from a fixed distribution. As a result, it struggles with out-of-distribution (OOD) generalization when faced with unseen automata. This issue is reflected in its performance degradation under more complex LTL specifications: the success rate $\eta_s$ drops from 0.9 on $\varphi_{1-4}$ to 0.82 on $\varphi_{5-8}$ and 0.94 on $\varphi_{9-12}$ to 0.7 on $\varphi_{13-16}$. Additionally, [54] has higher violation rates as it does not model the safety constraint explicitly.
>
> While it is possible to integrate automata embeddings with our method (e.g., via state augmentation), whether this will help mitigate the issue of myopic behavior requires further investigation, as it depends heavily on what the agent can observe (as we explain above), and conditioning a policy on information that is not observable can degrade its performance. Besides, integrating such embeddings introduces the additional challenge of OOD generalization (as we explain above and in Sections 5.1 and 5.2).
>
> |                   | ours $\eta_s$ | ours $\eta_v$ |    ours $\mu$    | RAD-embeddings $\eta_s$ | RAD-embeddings $\eta_v$ | RAD-embeddings $\mu$ |
> |:-----------------:|:-------------:|:-------------:|:----------------:|:-----------------------:|:-----------------------:|:--------------------:|
> |    $\varphi_1$    | $0.98\pm0.02$ | $0.00\pm0.00$ |   $7.38\pm0.13$  |      $0.96\pm0.04$      |      $0.00\pm0.00$      |    $18.71\pm2.06$    |
> |    $\varphi_2$    | $1.00\pm0.00$ | $0.00\pm0.00$ |   $5.48\pm0.09$  |      $0.90\pm0.07$      |      $0.07\pm0.04$      |    $13.64\pm2.71$    |
> |    $\varphi_3$    | $0.96\pm0.01$ | $0.00\pm0.00$ |   $8.61\pm0.46$  |      $0.86\pm0.05$      |      $0.08\pm0.03$      |    $20.02\pm2.05$    |
> |    $\varphi_4$    | $0.98\pm0.01$ | $0.00\pm0.00$ |   $7.41\pm0.37$  |      $0.89\pm0.04$      |      $0.00\pm0.00$      |    $18.78\pm2.82$    |
> |  $\varphi_{1-4}$  | $0.98\pm0.02$ | $0.00\pm0.00$ |   $7.22\pm1.18$  |      $0.90\pm0.06$      |      $0.04\pm0.04$      |    $17.79\pm3.37$    |
> |    $\varphi_5$    | $0.99\pm0.00$ | $0.00\pm0.00$ |  $10.60\pm0.12$  |      $0.88\pm0.05$      |      $0.00\pm0.00$      |    $26.02\pm1.62$    |
> |    $\varphi_6$    | $0.95\pm0.01$ | $0.00\pm0.00$ |   $7.48\pm0.06$  |      $0.90\pm0.04$      |      $0.00\pm0.00$      |    $19.48\pm1.96$    |
> |    $\varphi_7$    | $0.94\pm0.01$ | $0.00\pm0.00$ |  $10.97\pm0.08$  |      $0.73\pm0.05$      |      $0.17\pm0.05$      |    $26.07\pm3.73$    |
> |    $\varphi_8$    | $0.93\pm0.01$ | $0.00\pm0.00$ |  $10.24\pm0.08$  |      $0.76\pm0.11$      |      $0.13\pm0.05$      |    $25.60\pm2.89$    |
> |  $\varphi_{5-8}$  | $0.95\pm0.03$ | $0.00\pm0.00$ |   $9.82\pm1.42$  |      $0.82\pm0.10$      |      $0.07\pm0.08$      |    $24.29\pm3.77$    |
> |    $\varphi_9$    | $0.99\pm0.01$ | $0.01\pm0.01$ |  $380.15\pm5.17$ |      $0.90\pm0.07$      |      $0.08\pm0.07$      |   $423.11\pm51.08$   |
> |    $\varphi_{10}$   | $0.97\pm0.01$ | $0.02\pm0.02$ | $253.44\pm10.22$ |      $0.92\pm0.02$      |      $0.06\pm0.02$      |   $301.21\pm101.99$  |
> |    $\varphi_{11}$   | $0.99\pm0.01$ | $0.01\pm0.01$ | $249.57\pm11.85$ |      $0.93\pm0.06$      |      $0.04\pm0.02$      |   $247.83\pm42.98$   |
> |    $\varphi_{12}$   | $1.00\pm0.00$ | $0.00\pm0.00$ |  $135.60\pm8.06$ |      $1.00\pm0.01$      |      $0.00\pm0.00$      |   $105.97\pm17.31$   |
> |  $\varphi_{9-12}$ | $0.99\pm0.01$ | $0.01\pm0.01$ | $254.69\pm89.18$ |      $0.94\pm0.05$      |      $0.04\pm0.05$      |   $269.53\pm129.95$  |
> |    $\varphi_{13}$   | $0.98\pm0.02$ | $0.00\pm0.00$ | $436.25\pm18.35$ |      $0.69\pm0.17$      |      $0.00\pm0.00$      |   $664.59\pm29.91$   |
> |    $\varphi_{14}$   | $0.98\pm0.01$ | $0.00\pm0.00$ | $434.45\pm14.61$ |      $0.77\pm0.21$      |      $0.00\pm0.00$      |   $628.57\pm22.89$   |
> |    $\varphi_{15}$   | $0.97\pm0.01$ | $0.01\pm0.00$ | $388.01\pm15.47$ |      $0.62\pm0.09$      |      $0.02\pm0.01$      |    $680.11\pm9.51$   |
> |    $\varphi_{16}$   | $0.99\pm0.01$ | $0.00\pm0.00$ | $396.83\pm20.39$ |      $0.70\pm0.18$      |      $0.00\pm0.00$      |   $616.28\pm59.88$   |
> | $\varphi_{13-16}$ | $0.98\pm0.02$ | $0.01\pm0.01$ | $408.71\pm27.47$ |      $0.70\pm0.15$      |      $0.00\pm0.01$      |   $647.39\pm40.74$   |
>
> > 2. In Figure 6, what were the network sizes and number of training steps for "Ours (w/o s^\sigma)" and "Ours"? Were they the same? If so, then do you think that given more compute, "Ours (w/o s^\sigma)" could perform as well as "Ours"?
>
> For fair comparison, the network size and number of training steps for "Ours (w/o $s^\sigma$)" and "Ours" are the same. While it is possible that additional compute, such as more training steps or a larger network, could improve the performance of "Ours (w/o $s^\sigma$)", the underlying challenge remains. In the case of "w/o $s^\sigma$", the agent must learn over a much larger space that grows exponentially with the number of atomic propositions. This significantly increases sample complexity which is an issue that cannot be easily overcome by scaling compute alone.
>
> > 3. Does the benefit of the state reduction technique stem from the compression it provides through exploiting the symmetries in the observation? Is it possible to get the same performance with more compute when this state reduction technique is not used? I think a discussion or an experiment on this matter might be helpful to further highlight what this state reduction technique provides. Especially given the fact that it is an environment-specific approach that cannot be used in any environment, it would be good to have a further discussion of how to make the overall method work as well in cases where this state reduction technique cannot be used.
>
> The state reduction technique indeed exploits the symmetries in the obervation. When this technique is not applicable, we encode the subgoal using the bitvector encoding described in Section 4.1, and it still outperforms the baselines in terms of success and violation rates of the specifications, as shown in the ablation study in Section 5.3. However, it is important to note that due to the exponential complexity, i.e., $2^{|AP|}$, of the state-subgoal combinations (as discussed in lines 157-169), it would be very difficult to achieve comparable performance by simply adding compute when the number of atomic propositions is large.

---

> > ### Author Response · Authors · 2025-08-04
> > **Reminder for Re-evaluation and Further Discussion**
> >
> > Dear reviewer,
> >
> > Thanks again for your valuable comments. We hope our response has addressed your concerns, and we would greatly appreciate it if you could re-evaluate our submission based on the response, or let us know whether you have any other questions. Thanks!
> >
> > Best, \
> > Authors

---

> > ### Comment · Reviewer_VrZq · 2025-08-05
> >
> > I thank the authors for addressing my concerns and for providing the new experiment results. I recommend incorporating both the experiment results and the clarifications from the rebuttal into the final version of the paper.

---

> > > ### Author Response · Authors · 2025-08-05
> > >
> > > Thank you for your valuable review and feedback. We will revise our manuscript accordingly to incorporate the new experimental results into Section 5 and the clarifications on the observation reduction technique into Section 4.2 in the final version.

---

### Official Review · Reviewer_NmCA · 2025-07-02

**Clarity:** 1
**Significance:** 2
**Originality:** 1
**Rating:** 2
**Confidence:** 5

**Summary:**

This paper proposes GenZ-LTL, a reinforcement learning (RL) framework that enables zero-shot generalization to arbitrary Linear Temporal Logic (LTL) specifications. In its proposed method, the LTL task is first decomposed into a sequence of reach-avoid subgoals using the corresponding Büchi automaton. Instead of conditioning on full subgoal sequences, GenZ-LTL processes subgoals one at a time to better generalize to unseen tasks. Besides, the method treats avoid conditions as hard constraints and incorporates Hamilton-Jacobi reachability to enforce safety.

**Questions:**

Please address the novelty concerns I raised in Weakness section.

In the last paragraph of section 3, it says “q and q' are consecutive states along some accepting run, and q≠q' if q'∉ F”. Why q≠q' if q'∉ F? When the agent is going back and forth between two different accepting states, this statement is not correct.

	Authors use reach-avoid subgoals throughout the paper. Is this same as sub-task? If it is about reach-avoid, “sub-task” should be used, not subgoal.

	I don’t quite understand the observation reduction part. It says the original size of proposition-dependent part is k×2^(|AP|). The lidar observation has an independent component for every proposition. So, is the original size k×|AP|?

	What do you mean by this sentence “For simplicity of deriving a fusion operator that preserves task-relevant semantics such as distances to avoid zones, we assume that the os are coordinate-wise monotonic” in section 4.2?
	In section 5.3, authors evaluate the effects of the observation reduction and safe RL approach. However, training a separate Q and V function for safety constraints can cost a lot of environmental samples. Could you evaluate the increase of environmental samples used to train Q and V functions for safety constraints?

**Ethical Concerns:**

["NO or VERY MINOR ethics concerns only"]

**Final Justification:**

As I explained to authors during exchange of messages in rebuttal , and through my private message to the area chair, I do not see sufficient contribution to merit for publication. I stand my initial rating.

**Limitations:**

Authors need to include sufficient discussion about the scalability of the proposed method.

**Quality:**

1

**Strengths And Weaknesses:**

Strengths: This paper proposes to solve the generalization of temporal logic tasks, which is an important topic in many applications.

Weakness:
1. The major concern of this paper is its novelty. The main idea of this paper, i.e., solving or generalizing LTL tasks by decomposing LTL tasks into reach-avoid sub-tasks, has been proposed and investigated by many previous works [1,2,3,4]. Especially in [2,4], generalizing temporal logic tasks by training goal-conditioned RL policies to solving arbitrary reach-avoid sub-tasks has been proposed.

Authors say that safety constraints were not modeled in [4]. However, both target and safety constraints of sub-tasks/subgoals are clearly modeled in [2]. The method proposed in [2] also trains a general value function for arbitrary reach-avoid sub-tasks composed of propositions, but authors claim this as their innovation.

In [3], authors even derive the probability lower-bound for solving reach-avoid sub-tasks. Although [3] didn’t mention generalization, RAMS in [3] can be trivially applied into the generalization of LTL tasks.

In addition, the subgoal-induced observation reduction just trivially extracts components about propositions to reach and to avoid from LIDAR observations, since every proposition already has an independent component in the LIDAR observation.

2. The writing of this paper needs to be improved a lot.

The introduction part does not present some key concepts clearly, e.g., subgoal-induced observation reduction, which makes it difficult for people to understand the author's contribution.

In the method section, a concise algorithm table should be provided, summarizing key steps of the proposed framework. Authors only introduce the algorithm in text and diagram, but these are not enough. Besides, there are a lot of redundant sentences in the method section, making people confused when reading this part. In section 4.2, a diagram should be provided to describe the process of observation reduction.

3. Authors propose to use the constrained policy optimization [5] to solve the reach-avoid sub-task. However, there are many other safe RL approaches to achieve the same goal. Authors did not clearly explain why they chose the method in [5] and not use others. RASM in [3] is another method with probabilistic guarantee. Authors could compare their method with RASM for solving any reach-avoid sub-tasks.

[1] Wang, Yuning, and He Zhu. "Deductive Synthesis of Reinforcement Learning Agents for Infinite Horizon Tasks." 37th International Conference on Computer Aided Verification (CAV), 2025.

[2] Xu, Duo, and Faramarz Fekri. "Generalization of Compositional Tasks with Logical Specification via Implicit Planning." arXiv preprint arXiv:2410.09686 (2024).

[3] Žikelić, Đorđe, et al. "Compositional policy learning in stochastic control systems with formal guarantees." Advances in Neural Information Processing Systems 36 (2023): 47849-47873.

[4] Qiu, Wenjie, Wensen Mao, and He Zhu. "Instructing goal-conditioned reinforcement learning agents with temporal logic objectives." Advances in Neural Information Processing Systems 36 (2023): 39147-39175.

[5] Yu, Dongjie, et al. "Reachability constrained reinforcement learning." International conference on machine learning. PMLR, 2022.

---

> ### Author Rebuttal · Authors · 2025-07-30
>
> We appreciate the feedback from the reviewer. Below, we detail our responses to each of the weaknesses, questions, and limitations.
>
> ### Weaknesses
> >1.1. The major concern of this paper is its novelty, with additional points raised in comparison to prior work [1, 2, 3, 4].
>
> Our paper does not claim decomposing an LTL specification into reach-avoid subgoal sequences as a contribution. As discussed in lines 58–65, existing literature can be categorized into subgoal decomposition and direct specification encoding methods. Our work follows the subgoal decomposition paradigm, but as emphasized in lines 65–67 and 151–153, our contribution lies in how the reach-avoid subgoals are solved, which differs significantly from prior works.
>
> The main novelty and contributions of our paper, as stated in the introduction, are (1) showing that solving one subgoal at a time is in general better than conditioning on subgoal sequences, as done in the state-of-the-art method [DeepLTL]; (2) the first safe RL formulation with state-wise constraints for LTL satisfaction; and (3) subgoal-induced observation reduction for mitigating the exponential complexity of subgoal-state combinations.
>
> Regarding the treatment of safety constraints in prior works, we are very careful in our choice of language when describing them in our paper. In line 267, we state that "[4] does not model safety constraints explicitly within its RL formulation." [4] uses an ad hoc threshold over the value function to identify safe actions, which is not a principled mechanism for enforcing constraint satisfaction. [DeepLTL] uses a similar threshold-based technique, which is ineffective in achieving safety, as we discussed in lines 74-77 and Sections 5.1 and 5.2.
>
> [DeepLTL] Jackermeier and Abate. "DeepLTL: Learning to Efficiently Satisfy Complex LTL Specifications for Multi-Task RL," ICLR 2025 (oral).
>
> Regarding [2], although it takes as input the encoding of state, the reach and avoid subgoals (i.e., $p_{+}$ and $p_{-}$ in [2]), and a GNN-based embedding of the remaining sub-tasks, it still learns a single value function. As discussed earlier, using a single value function without explicit constraint modeling is insufficient for ensuring safety. In contrast, our approach is novel in how we solve the reach-avoid problem as we formulate the reach-avoid subgoals as constrained optimization problems and incorporate Hamilton-Jacobi (HJ) reachability for modeling safe actions explicitly (see Section 4.3). Such formulation contributes to the substantially lower violation rates observed in our results (see Section 5, especially the ablation study in Section 5.3). Besides, our version of HJ reachability also extends [5] to the LTL specification generalization setting, while [5] only considers a fixed task and safety constraint.
>
> We also respectfully disagree with the reviewer that [3] can be trivially applied to our setting. The probability lower bounds of [3] hinges on the following conditions:
> - Known Lipschitz constant of the system's dynamics.
> - Known reach (target) and avoid (unsafe) sets.
> - Full state space access and discretization.
> - An in-the-loop verifier, which, at each iteration, needs to perform interval bound propagation on neural networks and enumerate the discretized states.
>
> These assumptions and requirements are incompatible with our setting, where we can have partial observability, unknown dynamics and reach/avoid sets, and a high-dimensional and continuous state space that makes discretization intractable. Furthermore, [3] essentially uses PPO to learn the control policy and thus lacks a principled mechanism for enforcing hard safety constraints (similar to [2, 4, DeepLTL] that all learn a single value function).
>
> >1.2. "In addition, the subgoal-induced observation reduction just trivially extracts components ..."
>
> The actual implementation of the observation reduction may appear straightforward, as the fusion operator is environment/sensor-specific (see lines 190–193). However, the idea behind this technique is to circumvent the exponential complexity of state-subgoal combinations, a problem that has not received much attention in existing literature. As the number of atomic proposition grows, learning a goal-conditioned policy quickly becomes intractable. While various dimensionality reduction techniques have been proposed in the RL literature, they usually do not consider the state-subgoal combination (see lines 157-169). Our observation reduction provides a step toward more efficient policy learning to address this challenge. In the Conclusion section, we state that "We posit that the real complexity of LTL task generalization does not lie in the structure of the LTL formula or the corresponding automata, but rather in the combinatorial nature of the atomic propositions associated with the MDP states." We hope our work can inform and inspire future work in this direction.
>
> >2.1. “The introduction part does not present some key concepts clearly …”
>
> We thank the reviewer for pointing this out. We will add the following sentence in the introduction to convey the high-level idea of subgoal-induced observation reduction:
> "The idea of subgoal-induced observation reduction is to exploit symmetry in the state observations and extract information only relevant to the current subgoals."
>
> >2.2. “In the method section, a concise algorithm table should be provided, summarizing key steps of the proposed framework …”
>
> Due to the page limit, the formal algorithm and observation reduction details are placed in Appendix B/B.1. If space permits in a future version, we will move them to the main text.
>
> >3. Why did the authors choose [5] for safe RL, and why was RASM or other alternatives not considered for comparison?
>
> As stated in Section 4.3, "A key difference between our approach and existing subgoal-based approaches lies in the treatment of the avoid subgoal. We treat an avoid subgoal as a hard constraint, since whenever a violation occurs, satisfying the LTL formula becomes impossible...To enforce these hard constraints, we leverage the notion of HJ reachability as introduced in Section 3 and formulate our RL objective as follows." In Section 3 (lines 86-96), we introduce state-wise safety constraints and its connection to HJ reachability in the context of RL. Regarding [3], as we explain in our response to Weaknesses 1.1, RASM is not suitable for our setting due to its underlying assumptions and requirements.
>
> ### Questions
> >1. Why $q \neq q'$ if $q'\notin F$? When the agent is going back and forth between two different accepting states, this statement is not correct.
>
> We specify the condition $q \neq q'$ if $q' \notin F$ to filter out self-loops on non-accepting states in an accepting run. The case mentioned by the reviewer does not conflict with this condition. Our experiments on infinite-horizon tasks also include such case where the agent transitions back and forth between multiple accepting states (specifications listed in Table 4), and our method is able to satisfy those specifications effectively.
>
> >2. Terminology of reach-avoid subgoals or sub-tasks.
>
> We would like to clarify that we use the term reach-avoid subgoal consistently throughout the paper. While sub-task may also be appropriate, we chose subgoal to reflect the goal-conditioned nature of our policy and to maintain consistent terminology. In the literature, the two terms are used interchangeably; for example, [23, 50] use sub-task, and [39] uses subgoal to describe the same concept.
>
> >3. Size of the proposition-dependent part, $k \times 2^{|AP|}$or $k \times |AP|$?
>
> In general, the original size of the proposition-dependent part is $k\times 2^{|AP|}$ since the finite alphabet is $2^{AP}$ as noted in line 111. For instance, consider $AP$ = {blue, green}. The lidar observation may include two separate channels for the blue and green regions. However, these channels cannot cover the proposition blue $\land$ green (overlapping blue and green region).
>
> >4. What do you mean by this sentence “For simplicity of deriving a fusion operator ... we assume that the os are coordinate-wise monotonic” in section 4.2?
>
> The sentence refers to the difficulty of defining a general fusion operator that preserves meaningful task semantics (e.g., distance to avoid states) for arbitrary observation functions, which may be nonlinear and non-monotonic. Thus, for simplicity, we assume the observation functions are coordinate-wise monotonic. It is also possible to derive the fusion operator if we know what the observation functions are.
>
> >5. Training a separate Q and V function for safety constraints can cost a lot of environmental samples. Could you evaluate the increase of environmental samples used to train Q and V functions for safety constraints?
>
> During training, our method uses the same number of interactions (15M, as stated in Section 5) as all baselines. Each interaction yields a transition of the form $(s^\sigma_t, a_t, r_t, h_t)$, where $r_t$ and $h_t$ are used to learn the reward value function $V_r$ and the reachability value function $V_h$, respectively. Therefore, our method does not require additional environment samples.
>
> ### Limitations
> >1. Authors need to include sufficient discussion about the scalability of the proposed method.
>
> We would like to emphasize that we focus primarily on the zero-shot generalization of LTL specifications, which directly relates to scalability with respect to specification complexity. Our method scales linearly with the length of the subgoal sequences by completing the subgoals one at a time. In addition, our observation reduction technique improves scalability with respect to state/observation dimensions by extracting observations only relevant to the current subgoal.
>
> We hope that our responses sufficiently address the reviewer’s questions and concerns and kindly ask the reviewer to consider raising their score.

---

> > ### Author Response · Authors · 2025-08-04
> > **Reminder for Re-evaluation and Further Discussion**
> >
> > Dear reviewer,
> >
> > Thanks again for your valuable comments. We hope our response has addressed your concerns, and we would greatly appreciate it if you could re-evaluate our submission based on the response, or let us know whether you have any other questions. Thanks!
> >
> > Best,\
> > Authors

---

> > ### Comment · Reviewer_NmCA · 2025-08-06
> >
> > Thanks for your explanation. I will maintain my rating the same due to limited novelty of the work.

---

> ### Author Response · Authors · 2025-08-06
> **Response to Limited Novelty**
>
> We believe the reviewer is conflating task decomposition with task generalization. Existing literature on LTL task geneneralization is divided along several lines, e.g., automata/task graph embedding vs. task decomposition, and for the latter, conditioning on subtask/subgoal sequences vs. conditioning on individual subgoals. Task decomposition is just a subprocedure in one of the possible approaches to the problem. The common belief, including the latest SOTA (DeepLTL [ICLR2025]), is that conditioning on individual subgoals is myopic. Our paper is the first to show that conditioning on individual subgoals is actually better in general. This approach has the advantage of avoiding out-of-distribution (OOD), a core issue in generalization which arises when learning from subgoal sequences or automata embeddings. It enables training a goal-conditioned policy where the avoid subgoal is dealt with as a hard constraint (no other methods in the LTL task generalization literature does this, and some approaches simply cannot incorporate hard constraints). Additionally, our paper is the first to consider mitigating the exponential complexity of subgoal-state combinations. We hope the reviewer will also recognize the other merits of our paper, including the extensive experimental comparisons with all SOTA methods on LTL task generalization, showing that our paper outperforms them by a large margin in terms of both task satisfaction rate and performance.

---

> ### Comment · Reviewer_NmCA · 2025-08-06
>
> 1. The approach of zero-shot generalization using one subgoal or sub-task at a time, which is positioned as the primary contribution of submission 13551, is not a novel idea. Prior works [1–4] have already explored this concept extensively within the framework of LTL or its variants. Compared to these earlier efforts, the contribution of submission 13551 appears limited to incorporating avoidance constraints and solving a constrained RL problem via Hamilton-Jacobi (HJ) reachability—an approach that constitutes only an incremental advancement over existing methods.
>
> 2. Moreover, conditioning policy or value functions on a single subgoal/sub-task is generally sub-optimal, especially in environments containing multiple instances of similar objects. This limitation has been empirically demonstrated in recent literature, including a workshop paper [6], and is further supported by the findings in LTL2Action (ICML 2021). Hence, the claim that this conditioning strategy leads to effective generalization lacks solid justification.
> For completeness, the relevant prior works are listed below (also cited in the first review):
>
> 	•	[1] Wang, Yuning, and He Zhu. Deductive Synthesis of Reinforcement Learning Agents for Infinite Horizon Tasks. CAV 2025.
>
> 	•	[2] Xu, Duo, and Faramarz Fekri. Generalization of Compositional Tasks with Logical Specification via Implicit Planning. arXiv:2410.09686, 2024.
>
> 	•	[3] Žikelić, Đorđe, et al. Compositional Policy Learning in Stochastic Control Systems with Formal Guarantees. NeurIPS 2023.
>
> 	•	[4] Qiu, Wenjie, et al. Instructing Goal-Conditioned Reinforcement Learning Agents with Temporal Logic Objectives. NeurIPS 2023.
>
> 	•	[6] Giuri, Mattia, et al. Zero-Shot Instruction Following in RL via Structured LTL Representations. ICML 2025 Workshop.
>
>     3. Regarding the claimed innovation in observation space reduction: this is not innovative either. In the Zone environment based on Safety Gym, the observation space already consists of k × |AP| dimensions—k bins per atomic proposition—making further reduction unnecessary. For example, if the target is a conjunction of “blue” and “green,” the agent merely needs to attend to the direction where both attributes yield strong values. Importantly, no prior work used an observation space with k × 2^|AP| dimensions, as suggested by the authors.

---

> ### Author Response · Authors · 2025-08-07
> **Clarification on Contribution, Conditioning Strategy and Generalization**
>
> Regarding 1, our contribution lies precisely on how the subtasks/subgoals are solved. [4], which we empirically evaluate against in the paper, uses a heuristic to first find a subgoal sequence and then use an ad hoc threshold over the value function to solve the reach-avoid subgoals along that sequence. In contrast, our approach does not heuristically pick the subgoal sequence, but instead provides a mechanism for subgoal switching (see end of Section 4). More importantly, our approach solves the reach-avoid subgoals in a principled way by incorporating state-wise constraints when training the RL agent and solving them with a HJ-reachability-based method tailored for the LTL task generalization setting (no methods in the literature of LTL generalization, to our knowledge, have considered enforcing hard state-wise constraints since the problem was first considered in 2020). We would like to point out that what the reviewer perceives as "incremental advancement" produces huge leaps in performance over all SOTA methods in LTL generalization (see Table 1 and 2 in the paper).
>
> To further elaborate on [1-3] which we do not consider as SOTA in LTL task generalization for RL agents, [1] relies on strong assumptions similar to the RASM paper and is thus not applicable to our setting as explained earlier. It also handles only a fragment of LTL. [2], which is not peer-reviewed and contains limited evaluation, does not model the hard state-wise constraints, and is effectively a combination of subgoal conditioning and task graph conditioning (via state augmentation). It thus suffers from OOD issues which we will explain in our response to the reviewer's point 2 later. [3] is the RASM paper and we have explained why it does not apply to our setting due to its strong assumptions. We would like to additionally point out that due to these strong assumptions and the need of policy verification, they have only been shown to work on very small systems and environments.
>
> Regarding point 2, we respectfully disagree with the reviewer's assessment that "the claim that this conditioning strategy leads to effective generalization lacks solid justification" as we provide ample theoretical (from an OOD point of view) and experimental justification in the paper. While considering only one subgoal at a time can be suboptimal in certain cases, achieving globally optimal behavior across reach-avoid subgoal sequences **requires full observability of the environment and all the subgoal states**, a condition that is often unmet in practice and missed by existing literature (the exact condition is even stronger; value-based methods would require precise value estimates of all the subgoal states, which is not feasible in model-free and partially observable settings). Furthermore, we emphasize that our method is evaluated in environments where **all instances contain multiple occurrences of the same type of object** (e.g., multiple green regions) as visualized in Figure 11, 13 and 14 in the Appendix. On the other hand, current SOTA methods train policies conditioned on either automata embeddings or subgoal sequences, both of which suffer from OOD issue whereas our method does not (for further explanation, see our responses to Reviewer tAEC Weakness 1 and Reviewer VrZq Weaknesses and Questions 1). In addition, similar to all other methods in the literature of LTL task generalization, they do not incorporate hard constraints when training the RL agent (in fact, some methods cannot easily incorporate hard state-wise constraints). For additional evaluation results against SOTA automata-embedding based approach, see our rebuttal to reviewer VrZq. It is thus a new finding that conditioning on one subgoal at a time, contrary to popular belief, is better in general.
>
> Regarding point 3, we respectfully disagree with the reviewer's claim that reducing beyond k × |AP| is unnecessary, as our ablation study in Section 5.3 clearly shows substantial improvements when reducing k × |AP| based on the current subgoal in the ZoneEnv environments. We also respectfully disagree with the reviewer's claim that "if the target is a conjunction of “blue” and “green,” the agent merely needs to attend to the direction where both attributes yield strong values" and "no prior work used an observation space with k × 2^|AP| dimensions." Fundamentally, we have an alphabet of size 2^|AP| and ZoneEnv is just one specific environment. In the ZoneEnv case, we do not know the positions and the sizes of the color regions a priori (they are also randomized at test time). Each beam of the lidar observation only returns a value corresponding to the first (corresponding) color point that it sees along that line. Thus, a direction where both attributes yield strong values does not necessarily indicate an overlapping region (e.g., a blue region and then a non-overlapping green region some distance after it). Prior works have also considered lidar observation in 2^|AP| such as DeepLTL.

---

### Official Review · Reviewer_tAEC · 2025-07-03

**Clarity:** 3
**Significance:** 3
**Originality:** 3
**Rating:** 5
**Confidence:** 3

**Summary:**

This paper introduces GenZ-LTL, a reinforcement learning (RL) method that achieves zero-shot generalization to unseen tasks defined by Linear Temporal Logic (LTL) specifications. The core approach is to decompose LTL goals into reach-avoid subgoals through Büchi automata and subgoal-induced observation reduction, and solve them sequentially using a safe RL framework guided by Hamilton-Jacobi (HJ) reachability constraints. The subgoal-induced observation reduction technique also mitigate exponential scaling with atomic propositions. To cope with this issue when a selected subgoal is unsatisfiable in the environment, there is a timeout-based subgoal-switching mechanism to identify unsatisfiable subgoals and allow the agent to switch to alternative ones. Empirical results across several environments demonstrate generalization, safety, and sample efficiency compared to the baselines.

**Questions:**

1. Is there any comparison that highlights the performance of GenZ-LTL versus the baselines on satisfiable LTL specifications that may admit sub-optimal solutions?
2. How well does GenZ-LTL perform in scenarios that closely resemble real-world environment?
3. See above in Weaknesses

**Ethical Concerns:**

["NO or VERY MINOR ethics concerns only"]

**Final Justification:**

Most of my questions are resolved.

**Limitations:**

yes

**Paper Formatting Concerns:**

No formatting issues

**Quality:**

3

**Strengths And Weaknesses:**

Strengths:
1. First method to combine state-wise safety constraints, HJ reachability, with goal-conditional RL on LTL tasks.
2. The reach-avoid formulation and observation reduction technique enhance the generalization while keep the training efficiency.
3. Comprehensive evaluations across finite and infinite-horizon tasks, with strong baselines (DeepLTL, LTL2Action, GCRL-LTL)
4. The method and experiments are described clearly, with good figures and examples.

Weaknesses:
1. The method handles only one subgoal at a time, which could be suboptimal in tasks requiring global planning. The authors acknowledge this in their discussion, but it's still a core limitation.
2. The Assumptions for observation reduction(e.g., decomposability of observations, monotonic sensor fusion) might not generalize to all real-world settings.
3. It would be beneficial to include additional experimental results on benchmarks that closely resemble real-world scenarios and robot agents, such as robotic arms or quadrupeds in simulated environments in MuJoCo [1].

[1] MuJoCo benchmark: https://github.com/google-deepmind/mujoco

---

> ### Author Rebuttal · Authors · 2025-07-30
>
> We thank the reviewer for their thoughtful comments and constructive feedback. We address each weakness and question below.
>
> ## Weaknesses
> > 1. The method handles only one subgoal at a time, which could be suboptimal in tasks requiring global planning. The authors acknowledge this in their discussion, but it's still a core limitation.
>
> Considering only one subgoal at a time indeed can be suboptimal. However, achieving globally optimal behavior across reach-avoid subgoal sequences requires full observability of the environment and all the subgoal states, a condition that is often unmet in practice. In addition, our goal is to zero-shot generalize to *arbitrary* LTL specifications, where their subgoal sequences may have very long length, making global planning intractable.
>
> Thus, in this paper, we argue that it is actually better to consider one subgoal at a time in general. This approach has the advantage of avoiding out-of-distribution (OOD) generalization issues that arise when learning from subgoal sequences -- a limitation observed in the DeepLTL baseline (see Section 5.1 and 5.2, and Appendix C where DeepLTL's performance drops significantly under partial observability). In addition, conditioning a goal-conditioned policy on individual subgoals rather than subgoal sequences or automaton embeddings reduces the input dimensionality, thus improving sample efficiency. Lastly, it enables training a goal-conditioned policy where the avoid subgoal can be dealt with as a hard constraint.
>
> We also provide additional evaluation results of RAD-embeddings [54], which can be viewed as having knowledge of the global plan as it learns policies conditioned on embeddings of the automata. As shown in our response to Reviewer VrZq (Weakness 1), our approach substantially outperforms RAD-embeddings in terms of both specification satisfaction rate and performance.
>
> > 2. The Assumptions for observation reduction (e.g., decomposability of observations, monotonic sensor fusion) might not generalize to all real-world settings.
>
> We acknowledge that the assumptions underlying observation reduction may not generalize to all real-world settings. However, we would like to emphasize that the core idea behind this technique is broadly applicable. When solving the reach-avoid problem, what truly matters is whether a state satisfies the reach or avoid condition under the current subgoal, rather than its specific label $L(s)$ as we mentioned in Section 4.2. Moreover, our method can be extended to more realistic settings with some additional effort, but without requiring fundamental changes to the core approach. For example, in autonomous driving scenarios with multiple sensors such as cameras and LiDAR, observation reduction can be enabled through perception modules. Semantic segmentation can be applied to both image and point cloud data to extract relevant features based on the current reach-avoid subgoals. For instance, if the avoid subgoal involves avoiding other cars and pedestrians, and the reach subgoal involves staying within the current lane, then perception can be used to segment the scene accordingly. This allows the agent to focus on task-relevant information and use processed observations, rather than raw sensory data, to learn value functions and policies effectively.
>
> > 3. It would be beneficial to include additional experimental results on benchmarks that closely resemble real-world scenarios and robot agents, such as robotic arms or quadrupeds in simulated environments in MuJoCo.
>
> We agree with the reviewer that additional experiments on robotic benchmarks could further strengthen the paper. We would like to point out that the notion of generalization spans multiple dimensions, such as unseen task specifications, novel environments, and varying robot platforms. This work focuses specifically on generalization to unseen task specifications. The environments used in our work are widely adopted in prior studies, such as [23, 39, 50], and were chosen to ensure fair and meaningful comparisons. While MuJoCo-based environments are valuable for studying complex dynamics, they are less suited for evaluating generalization to temporally extended or logic-based tasks. Nonetheless, we agree that broader benchmark coverage is important, and we would be enthusiastic to evaluate our method on future benchmarks that reflect the challenges of LTL task generalization.
>
> ## Questions
> > 1. Is there any comparison that highlights the performance of GenZ-LTL versus the baselines on satisfiable LTL specifications that may admit sub-optimal solutions?
>
> As shown in Table 1, for some specifications such as $\varphi_4$ and $\varphi_{9-12}$, the average number of steps $\mu$ among successful trajectories is higher for our method (i.e. suboptimal) than for DeepLTL (a method that considers subgoal sequences). However, our method achieves a higher success rate $\eta_s$ and a lower violation rate $\eta_v$ across these specifications. As discussed in lines 279–284, $\mu$ is a secondary metric as step count is less important when the agent struggles to satisfy the specifications. Also, our method achieves lesser number of steps (i.e. better performance) in all the other specifications in the paper.
>
> > 2. How well does GenZ-LTL perform in scenarios that closely resemble real-world environment?
>
> ZoneEnv is representative of real-world navigation environments where tasks involve spatial, temporal, and logical constraints, thus suitable for evaluating LTL task generalization. It also features a high-dimensional state space (e.g., LiDAR observations) and realistic robot dynamics (e.g., differential-drive robots), providing a challenging testbed. As noted in our response to Weakness 1, we would be enthusiastic to evaluate our method in future benchmarks that reflect the challenges of LTL task generalization. For further discussion on how our method applies to broader applications, please see our responses to Weaknesses 2 and 3.

---

> > ### Author Response · Authors · 2025-08-04
> > **Reminder for Re-evaluation and Further Discussion**
> >
> > Dear reviewer,
> >
> > Thanks again for your valuable comments. We hope our response has addressed your concerns, and we would greatly appreciate it if you could re-evaluate our submission based on the response, or let us know whether you have any other questions. Thanks!
> >
> > Best, \
> > Authors

---

> > > ### Comment · Reviewer_tAEC · 2025-08-06
> > >
> > > Thank you answering my questions. While I wished there could be evaluating on more complex benchmarks like MuJoCo.
> > > > While MuJoCo-based environments are valuable for studying complex dynamics, they are less suited for evaluating generalization to temporally extended or logic-based tasks.
> > > I can't agree with this statement because I have seen papers evaluating their methods on mujoco which have demonstrated pretty good performance. I really want to know how robustness this work is. Since the lack of experimental result on this part, I would like to remain current rating.

---

> > > > ### Author Response · Authors · 2025-08-06
> > > >
> > > > Thank you for your feedback. If possible, could you share the reference to the papers you have in mind so that we could evaluate our method and the baselines in those environments? In the meantime, we will also modify the original ZoneEnv by replacing its dynamics with a more complex agent such as Ant to further evaluate the performance of our method.

---

> ### Author Response · Authors · 2025-08-07
> **Additional Experiments with Complex Dynamics**
>
> To evaluate the performance of our method in environments with more complex dynamics, we have conducted additional experiments using the Ant agent (Safety Gymnasium [NeurIPS 2023]), consisting of a torso and four legs connected by hinge joints, in the Zone environment. The state space includes the agent’s ego-state (40 dimensions, compared to 12 for the Point agent used in our main experiments) and LiDAR observations of the regions. The action space has 8 dimensions, corresponding to the torques applied to the Ant’s joints to coordinate leg movements (compared to 2 dimensions for the Point agent). We train and evaluate our method against DeepLTL, the strongest baseline in our main results. For a fair comparison, all training procedures, training parameters, and neural network sizes were kept the same (note that the last one in particular could result in some performance drop as the state dimension and action dimension are bigger for the ant agent). The evaluation results of 5 seeds and 100 trajectories per seed are shown below. We report the success rate $\nu_s$, violation rate $\nu_v$, and average number of steps $\mu$ to satisfy the specification among successful trajectories. We can observe that our method consistently outperforms the baseline across all metrics, demonstrating an even larger performance gain in environments with more complex agent dynamics. Note that for the Ant agent, in addition to the LTL specifications, there is a built-in safety constraint that the agent should not fall headfirst, enforced as a hard constraint with immediate episode termination. For DeepLTL, a negative reward is assigned in such cases, similar to how it is done when a specification is violated, as opposed to handling the safety constraint through HJ reachability in our approach. This impedes the learning of coordinated locomotion and then the satisfaction of LTL specifications, leading to substantially degraded performances in the case of DeepLTL.
>
> |                   | ours $\eta_s$ | ours $\eta_v$ |    ours $\mu$    | DeepLTL $\eta_s$ | DeepLTL $\eta_v$ |   DeepLTL $\mu$   |
> |:-----------------:|:-------------:|:-------------:|:----------------:|:----------------:|:----------------:|:-----------------:|
> |    $\varphi_9$    | $0.97\pm0.02$ | $0.02\pm0.01$ | $305.32\pm46.50$ |   $0.00\pm0.01$  |   $0.23\pm0.08$  |  $865.50\pm72.83$ |
> |   $\varphi_{10}$  | $0.91\pm0.03$ | $0.09\pm0.03$ | $202.00\pm53.33$ |   $0.03\pm0.02$  |   $0.39\pm0.11$  | $588.58\pm166.33$ |
> |   $\varphi_{11}$  | $0.95\pm0.02$ | $0.04\pm0.02$ | $182.88\pm38.66$ |   $0.07\pm0.02$  |   $0.27\pm0.05$  |  $604.87\pm28.22$ |
> |   $\varphi_{12}$  | $0.98\pm0.01$ | $0.00\pm0.00$ | $125.97\pm35.66$ |   $0.24\pm0.05$  |   $0.05\pm0.04$  |  $392.04\pm64.39$ |
> |  $\varphi_{9-12}$ | $0.95\pm0.04$ | $0.04\pm0.04$ | $204.04\pm77.69$ |   $0.09\pm0.10$  |   $0.23\pm0.14$  | $568.15\pm172.57$ |
> |   $\varphi_{13}$  | $0.93\pm0.04$ | $0.00\pm0.00$ | $459.99\pm86.69$ |   $0.00\pm0.00$  |   $0.00\pm0.00$  |         -         |
> |   $\varphi_{14}$  | $0.95\pm0.02$ | $0.00\pm0.00$ | $434.41\pm76.87$ |   $0.00\pm0.00$  |   $0.00\pm0.00$  |         -         |
> |   $\varphi_{15}$  | $0.89\pm0.02$ | $0.08\pm0.02$ | $429.76\pm75.75$ |   $0.00\pm0.00$  |   $0.39\pm0.13$  |         -         |
> |   $\varphi_{16}$  | $0.95\pm0.03$ | $0.00\pm0.00$ | $449.89\pm83.26$ |   $0.00\pm0.00$  |   $0.00\pm0.00$  |         -         |
> | $\varphi_{13-16}$ | $0.93\pm0.04$ | $0.02\pm0.04$ | $443.51\pm73.31$ |   $0.00\pm0.00$  |   $0.10\pm0.18$  |         -         |

---

> ### Author Response · Authors · 2025-08-09
> **Reminder for Re-evaluation and Further Discussion**
>
> Dear Reviewer,
>
> Thank you again for your valuable feedback. We hope our clarifications and additional experiments have addressed your concerns. Given that the discussion period will conclude in the next 12 hours, we would greatly appreciate it if you could consider raising your score if our response has resolved your concerns, or let us know if you have any remaining questions.
>
> Best,
> The Authors

---

### Comment · Area_Chair_ferh · 2025-08-05

Dear Reviewers,

Thank you again for your thoughtful reviews of paper 13551.

The authors have submitted detailed rebuttals addressing your major concerns. As the discussion phase is ending soon, please take a moment to review the responses and indicate whether your concerns have been resolved, or provide follow-up comments if further clarification is needed.

Best regards,
Your AC

---

### Note · Authors · 2025-08-12

We thank the chairs and reviewers for their time, feedback, and dedication to a rigorous review process. We are glad that most reviewers recognized our novel contributions and comprehensive evaluation, which show that our method substantially outperforms existing approaches in learning zero-shot generalizable policies. Below, we reiterate the key points of our responses.

- Task decomposition should not be conflated with task generalization - decomposing an LTL task specification into subtasks and then training a policy for each subtask alone is not adequate for generalization as the trained policies do not readily generalize to novel tasks. Thus, in addition to its strong assumptions noted in the rebuttal, the RASM paper (repeatedly brought up by Reviewer NmCA) is not applicable to zero-shot generalization as it needs to re-train and re-verify a policy when given a new specification (or when the reach/avoid set changes).
- A popular belief in existing literature is that solving one subgoal at a time is myopic. Latest SOTAs have focused on giving the agent more foresight, such as training a policy based on embeddings of the whole specification automata or subgoal sequences. A main contribution of our paper is that we show the opposite - solving one subgoal at a time is better for zero-shot generalization. This is based on several key observations and ideas: (1) achieving globally optimal behavior across subgoal sequences requires full observability and precise value estimates of the environment and subgoal states, conditions that are often unmet in practice, (2) conditioning on automata or subgoal sequences is prone to out-of-distribution issues, as one can always give the agent a more complex LTL specification, (3) the avoid subgoals must be enforced as hard constraints during policy training, a critical aspect overlooked in prior work, and (4) we only need to focus on observations relevant to the current subgoal, thus reducing the state space for the RL agent.
- We present extensive experiments on test-time randomized environments, demonstrating significant improvements (in both task satisfaction and performance) over prior SOTAs (LTL2Action [ICML2021], GCRL-LTL [NeurIPS2023], RAD-embeddings [NeurIPS2024], and DeepLTL [ICLR2025(oral)]). In addition, we provide additional results on the MuJoCo Ant agent (40-dimensional ego-state), demonstrating that our method performs well in settings with both complex agent dynamics and complex LTL task specifications.

---

### Decision · Program_Chairs · 2025-09-17

**Decision:**

Accept (poster)

**Comment:**

This paper presents GenZ-LTL, a framework for zero-shot generalization to LTL tasks by sequentially solving reach-avoid subgoals with state-wise constraints and observation reduction. The method is clearly described and empirically validated, showing consistent improvements over baselines. While the contributions are somewhat incremental and the framing as a safe RL method may be overstated, the work still offers a useful contribution to specification-guided RL.